# Simultaneously Perturbed Optimistic Gradient Methods for Payoff-Based Learning in Games

## Abstract

We examine the long-run behavior of learning in a repeated game where the agents operate in a low-information environment, only observing their realized payoffs at each stage. We study this problem in the context of monotone games with unconstrained action spaces, where standard gradient schemes may lead to cycles, even with perfect gradient information. To account for the fact that only a *single* payoff observation can be made at each iteration—and no gradient information is directly observable—we design and deploy a simultaneous perturbation gradient estimation method adapted to the challenges to the problem at hand, namely unbounded action spaces, gradients and rewards. In contrast to single-timescale approaches, we find that a two-timescale approach is much more effective at controlling the (unbounded) noise introduced by payoff-based gradient estimators in this setting. Owing to the introduction of a second timescale, we show that the proposed simultaneously perturbed optimistic (SPOG) algorithm converges to equilibrium with probability 1. In addition, by developing a new method to assess the rate of convergence of two-timescales stochastic approximation procedures, we show the sequence of play induced by SPOG converges at an asymptotic $\tilde{\mathcal{O}}(n^{-2/3})$ rate in strongly monotone games. To the the best of our knowledge, this is the first convergence rate result for games with unbounded action spaces, and it is faster than the sharpest known convergence rates for single-observation, payoff-based learning in strongly monotone games with bounded action spaces.

## 1 Introduction

Many large-scale systems involve the interaction of multiple autonomous decision makers. Examples of this include generative adversarial networks (GANs) (Goodfellow et al., 2014), distributed optimization in parallel computing, transportation networks (Vigneri et al., 2019), etc. In this setting, each agent must respond to the changing environment posed by the other agents' actions, and the utility of each agent is determined by the actions of all players through a fixed underlying rule.

In a game-theoretical setting, first-order gradient methods might never stabilize in the long-run, resulting in cycles or even divergence of the sequence of play, even in simple, unconstrained bilinear min-max games (Daskalakis et al.). In fact, even optimistic gradient (OG) methods, which incorporate a recency bias, have been shown to exhibit trajectories of play that orbit an equilibrium, failing to converge when feedback is contaminated with noise (Hsieh et al., 2022). An example of this is illustrated in Figure 2d.

When each agent has a noise-contaminated estimate of their payoff gradient, a modification of the optimistic gradient method, known as OG+, results in last-iterate convergence of the sequence of play to a Nash equilibrium in monotone games (Hsieh et al., 2020; 2022). This modification involves a *learning rate separation*, whereby the extrapolation step is taken with a learning rate an order of magnitude larger than that of the update step. By following a policy that explores aggressively and updates conservatively, the gradient noise effectively becomes an order of magnitude smaller than the expected variation of payoffs, ultimately enabling convergence.

In a low-information environment, an agent might not have access to an estimate of their gradient. We instead consider that the only information available to each agent is the payoff they receive at

| Algorithm | Actions | Monotone | Feedback | Convergence Rate | Type |
|---|---|---|---|---|---|
| AOG (Cai and Zheng, 2023) | Compact | Mere | Perfect FO | $\mathcal{O}(n^{-1})$ GAP | anytime |
| Dong et al. (2025) | Compact | Mere | 1-Point ZO | $\mathcal{O}(n^{-1/4}) \left\|\cdot\right\|^2$ | asymptotic |
| MD (Bravo et al., 2018) | Compact | Strong | 1-Point ZO | $\mathcal{O}(n^{-1/3}) \left\|\cdot\right\|^2$ | asymptotic |
| MD (Drusvyatskiy et al., 2022) | Compact | Strong | 1-Point ZO | $\mathcal{O}(n^{-1/2}) \left\|\cdot\right\|^2$ | anytime |
| Tatarenko and Kamgarpour (2024a) | Compact | Strong VS | 1-Point ZO | $\mathcal{O}(n^{-1/2}) \left\|\cdot\right\|^2$ | asymptotic |
| | | | 2-Point ZO | $\mathcal{O}(n^{-1}) \left\|\cdot\right\|^2$ | asymptotic |
| GABP (Abe et al., 2025) | Compact | Mere | Perfect FO | $\tilde{\mathcal{O}}(n^{-1})$ GAP | anytime |
| | | | Stoch. FO | $\tilde{\mathcal{O}}(n^{-1/7})$ GAP | anytime |
| **SPOG** | Unbounded | Strong | 1-Point ZO | $\tilde{\mathcal{O}}(n^{-2/3}) \left\|\cdot\right\|^2$ | asymptotic |

Table 1: Rates of convergence for learning algorithms in monotone games. See Sections 2.1-2.3.

each stage of the game. A difficulty inherent to this payoff-based context is that agents must estimate their payoff gradient from a *single* payoff observation. Despite these difficulties, no-regret, *payoff-based* learning algorithms have been developed that guarantee last-iterate convergence in monotone games with constrained action spaces (Bravo et al., 2018; Tatarenko and Kamgarpour, 2024a).

Further challenges arise when learning in games with unbounded action spaces. Standard compactness arguments cannot be applied to establish the convergence of iterates. Furthermore concave payoff functions are in general unbounded and not Lipschitz, adding a further layer of variance to zeroth-order (ZO) gradient estimators.

**Our contributions in the context of related work.**

1. We develop a simultaneously perturbed optimistic gradient (SPOG) learning algorithm that combines the learning rate separation technique present from OG+ with a novel, thresholded single-observation payoff-based gradient estimator. We show that SPOG converges to a Nash equilibrium with probability 1 in a large class of monotone games.

2. We obtain an asymptotic *last-iterate* rate of convergence of rate of $\tilde{\mathcal{O}}(n^{-2/3})$ in strongly monotone games. This is, to the best of our knowledge, the first rate of convergence result for unconstrained games and exceeds the corresponding best rate for strongly monotone *constrained* games with one-point ZO feedback Tatarenko and Kamgarpour (2024a). By reusing previous payoff observations as a baseline reward in the gradient estimate, thereby reducing the variance, our algorithm exceeds the sharpest known convergence rate for one-point ZO algorithms (Shamir, 2013; Ba et al., 2025).

3. We develop and deploy a new analysis method for two-timescales stochastic approximation (Borkar, 1997; Doan, 2021) to control the convergence rate of our algorithm.

Closely related work, summarized in Table 1, is described in Section 2.3 once we have introduced the necessary preliminaries.

## 2 PRELIMINARIES

### 2.1 MONOTONE GAMES IN NORMAL FORM

We consider games with a finite number $N$ of players and unconstrained continuous action spaces. Denote the set of players as $\mathcal{N} = \{1, \dots, N\}$. During play, each player $i \in \mathcal{N}$ simultaneously selects an *action* $x_i$ from their action set $\mathcal{X}_i = \mathbb{R}^{D_i}$, resulting in a *joint action profile* $x = (x_i, x_{-i}) \equiv (x_1, \dots, x_N) \in \mathcal{X} \equiv \prod_{i \in \mathcal{N}} \mathcal{X}_i$. Each player receives a *reward*, with Player $i$ receiving $u_i(x_i, x_{-i})$, where $u_i : \mathcal{X} \to \mathbb{R}$ is Player $i$'s *utility* function. Such a game is referred to as a *continuous game in normal form*. Write $v_i(x) := \nabla_{x_i} u_i(x_i; x_{-i})$ for the players' individual payoff gradients and define the game's *pseudo-gradient* operator $v : \mathbb{R}^D \to \mathbb{R}^D$ as $v(x) = (v_i(x))_{i \in \mathcal{N}}$ for all $x \in \mathcal{X}$.

**Definition 2.1** (Monotonicity). For $\mu \geq 0$, a map $v : \mathcal{X} \to \mathcal{Y}$ is said to be $\mu$-*monotone* over $\mathcal{X} \subset \mathbb{R}^D$ if the following inequality holds for all $x, x' \in \mathcal{X}$,

$$\langle v(x) - v(x'), x - x' \rangle \leq -\mu \left\| x - x' \right\|^2, \tag{MON}$$

where $\|\cdot\|$ denotes the Euclidean norm. If $\mu > 0$ then $v$ is *strongly monotone*, otherwise $v$ is *merely monotone*. A game is said to be (strongly/merely) *monotone* if its pseudo-gradient $v$ is (strongly/merely) *monotone* over its joint action space $\mathcal{X}$.

Monotonicity has thus given rise to a rich class of games, containing all bilinear min-max games (an unconstrained analogue of finite two-player, zero-sum games), games that admit a concave potential, and is common in applications to generative models (Chavdarova et al., 2019; Kamalaruban et al., 2020). Throughout the rest of this paper, we restrict our study to *monotone* games.

## 2.2 SOLUTION CONCEPTS

A widespread solution concept in the theory of games is the *Nash equilibrium* (Nash, 1951), a joint strategy profile from which no player can profit from deviating unilaterally. Formally,

**Definition 2.2** (Nash Equilibrium). An action profile $x^\star \in \mathcal{X}$ is said to be a *Nash equilibrium* if

$$u_i(x_i^\star; x_{-i}^\star) \geq u_i(x_i; x_{-i}^\star) \quad \text{for all } x_i \in \mathcal{X}_i, i \in \mathcal{N}. \tag{NE}$$

Let $\mathcal{X}_\star$ denote the set of Nash equilibria of the game. By concavity of the players' payoff functions, $\mathcal{X}_\star$ coincides exactly with the zeros of the pseudo-gradient $v$, i.e. $\mathcal{X}_\star = \{x \in \mathcal{X} : v(x) = 0\}$.

Throughout we impose the following assumptions on the underlying game.

**Assumption 1.** There exists constants $L > 0$, $\mu \geq 0$ such that

   (i) $v$ is $L$-Lipschitz, that is, $\|v(x) - v(x')\| \leq L \|x - x'\|$ for all $x, x' \in \mathcal{X}$;

  (ii) the game $\mathcal{G}$ is $\mu$-monotone;

 (iii) the set $\mathcal{X}_\star$ is non-empty;

 (iv) there exists a constant $G > 0$ satisfying $\|\nabla u_i(x)\| \leq G(1 + \|x\|)$ for all $x \in \mathcal{X}$.

Much of our analysis exploits the smoothness from Assumption 1(i) with iterative application of the monotonicity from Assumption 1(ii) to obtain last-iterate convergence guarantees. Our study concerns games with unconstrained action spaces, where in general Nash equilibria might not even exist; we avoid this difficult by imposing Assumption 1(iii) which is a standard assumption in this setting (Hsieh et al., 2022). The regularity Assumption 1(iv) is used to control the variance of our zeroth-order gradient estimate, which is necessary since the action spaces are unbounded.

## 2.3 RELATED WORK

In this section we provide a detailed account of the related work, as summarized in Table 1. We distinguish between first order (FO) and zeroth-order (ZO) feedback schemes.

**First-Order Feedback.** Much of the literature on online learning in games assumes that players are able to obtain gradient information by querying a *first-order* oracle (Nesterov, 2013), that is a "black-box" feedback scheme that returns an estimate $\hat{v}_i$ of Player $i$'s individual payoff gradient $v_i(x)$ at the current (joint) action profile $x = (x_i, x_{-i}) \in \mathcal{X}$. The oracle might be *perfect*, yielding $\hat{v}_i = v_i(x)$, or *stochastic* where the gradient is contaminated with some noise $U_i$.

In constrained games with mere monotonicity ($\mu \geq 0$), Abe et al. (2025) develop a payoff perturbation technique enabling last-iterate convergence at anytime rates of $\tilde{O}(n^{-1/7})$ for zero-mean bounded-variance additive noise, and at $\tilde{O}(n^{-1})$ with perfect gradient feedback. In the noiseless setting, the Accelerated Optimistic Gradient (AOG) converges with an optimal anytime rate $\mathcal{O}(n^{-1})$ (Cai and Zheng, 2023).

**Zeroth-Order Feedback.** In our study we consider instead *zeroth-order*, or *payoff-based*, feedback. In this setting, the only information available to agent $i \in \mathcal{N}$ is the actual payoff $u_i(x_i, x_{-i})$ that they receive at each stage of the game. Each agent is unaware of the payoff received by other agents, the actions of other agents, or even the number of agents in the game.

In this setting, agents must estimate their individual payoff gradient using only their observed payoff. Multi-point directional sampling techniques are an effective way to estimate a function's gradient (Kiefer and Wolfowitz, 1952; Flaxman et al., 2004), but require multiple queries of their payoff function. In general, this is not possible in a game theoretical setting, where a player's individual

payoff function might depend on the actions of *all* players, changing from one instance to the next as a result of the actions of other players.

Fortunately, techniques for estimating a function's gradient from a single function evaluation exist; most notably *simultaneous perturbation stochastic approximation* (SPSA) (Spall, 1997; Flaxman et al., 2004), which we define in Section 2.4. In a game theoretical setting, online learning algorithms using SPSA (or similar) gradient estimators have yielded last-iterate convergence results in games with *constrained action spaces* (Bravo et al., 2018; Tatarenko and Kamgarpour, 2024b). Bravo et al. (2018) develop a variant of mirror descent (MD) which they show enjoys an asymptotic last-iterate convergence rate of $\mathcal{O}(n^{-1/3})$ in *strongly monotone* games. Tighter analysis by Drusvyatskiy et al. (2022) reveals that this algorithm converges at an anytime rate of $\mathcal{O}(n^{-1/2})$, matching Tatarenko and Kamgarpour (2024a) who obtain this rate asymptotically in constrained games with a *strongly variationally stable* (VS) Nash equilibrium, a large class containing strongly monotone games. In games which are *merely monotone*, Dong et al. (2025) develop a doubly regularized variant of mirror descent that converges at an asymptotic rate of $\mathcal{O}(n^{-1/4})$, however, to achieve this rate, their algorithm requires a game-dependent choice of regularizer. Interestingly, we see the same rate appearing in the lower bound proved by Fiegel et al. (2025) for two-player zero-sum matrix games, which are *a fortiori* merely monotone.

Unlike all of the works above where action spaces are assumed to be *compact*, we consider the problem of payoff-based learning in *unconstrained* monotone games. In this setting, there do not seem to be any theoretical convergence rate guarantees in the literature.

## 2.4 THE SPSA GRADIENT ESTIMATOR

We define the SPSA gradient estimator of Spall (1997) in detail. Suppose that players are estimating $v(z)$ at joint action profile $z = (z_1, \ldots, z_N)$. For a *query radius* $\delta > 0$, each player $i \in \mathcal{N}$,

1. Samples a vector $w_i$ from the unit sphere $\mathbb{S}^{D_i} \subset \mathbb{R}^{D_i}$ and plays $\tilde{z}_i = z_i + \delta w_i$.

2. Receives feedback $\hat{u}_i = u_i(\tilde{z}_i, \tilde{z}_{-i})$ and constructs the estimate $\hat{V}_i = \frac{D_i}{\delta} \hat{u}_i w_i$.

As demonstrated in (Flaxman et al., 2004; Bravo et al., 2018), $\hat{V}_i$ is an unbiased estimator of the gradient of a $\delta$-*smoothing* $u_i^\delta(z)$ of $u_i$ evaluated at $z$. In particular, $\|\nabla_i u_i - \nabla_i u_i^\delta\|_\infty = \mathcal{O}(\delta)$ and the variance of $\hat{V}_i$ is $\mathcal{O}(\delta_n^{-2})$. Since $\|\hat{V}_i\|$ is proportional to the payoff received, and concave payoff functions on *unbounded* action spaces are unbounded, we must contend with the fact that the variance of this estimator may explode. An important observation is that, for any predetermined $c_i \in \mathbb{R}$, the *adjusted* SPSA (SPSA+) estimate

$$V_i = \frac{D_i}{\delta}(\hat{u}_i - c_i)w_i \qquad \text{(SPSA+)}$$

has the same expectation as $\hat{V}_n$. A key idea is that if $c_i = u_i(\tilde{z}^-)$ is chosen to be a *previous* payoff observation, and $\|\tilde{z} - \tilde{z}^-\|$ is sufficiently small, then for $u_i$ satisfying the regularity Assumption 1(iv), the exploding variance from the factor of $\delta_n^{-2}$ can be controlled.

We will find it useful to express SPSA+ in the form $V_i = v_i(z) + \xi_i$ where $\xi_i = U_i + b_i$ with unbiased component $U_i = V_i - \mathbb{E}V_i$ and bias $b_i = \nabla_i u_i^\delta(z) - \nabla_i u_i(z)$.

## 2.5 OPTIMISTIC GRADIENT METHODS AND THE ROLE OF LEARNING RATE SEPARATION

It is well known that simply following a gradient ascent/descent policy can result in non-convergence in games. Itself a variant of the *extragradient* (EG) algorithm (Korpelevich, 1976), the *optimistic gradient* (OG) algorithm (Popov, 1980; Rakhlin and Sridharan, 2013) mitigates non-convergence phenomena in online learning in games. Agents prescribed to an OG scheme use past gradient information to make an informed "look-ahead", extrapolation step, before taking an update step to update their strategy, as illustrated in Figure 1. Formally, assuming for the moment that players may query a stochastic *first-order* oracle $\tilde{v}$, the OG algorithm is defined by the sequence of iterates

$$\begin{aligned} X_{n+1/2,i} &= X_{n,i} + \gamma_{n-1}\tilde{v}_i(X_{n-1/2}), \\ X_{n+1,i} &= X_{n,i} + \gamma_n\tilde{v}_i(X_{n+1/2}), \end{aligned} \qquad \text{(OG)}$$

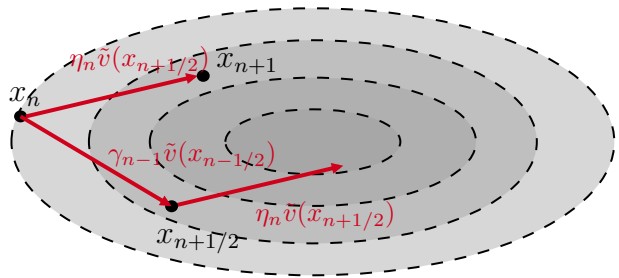

Figure 1: Optimistic Gradient with learning rate separation

where $\gamma_n > 0$ is a sequence of learning rates.

This technique stabilizes the learning dynamics when the agents have access to a *perfect* first-order oracle, yet can lead to divergent trajectories of play with a *stochastic* first-order oracle, even for simple two-player bilinear min-max games (Hsieh et al., 2022), illustrated in Figure 2d. To overcome this difficulty, Hsieh et al. (2022) introduce a *learning-rate separation* technique, which they term OG+, whereby the extrapolation step is taken with a learning rate that is asymptotically larger than that of the update step. Formally, the OG+ algorithm is defined by the sequence of iterates

$$\begin{aligned}
X_{n+1/2,i} &= X_{n,i} + \gamma_{n-1}\tilde{v}_i(X_{n-1/2}), \\
X_{n+1,i} &= X_{n,i} + \eta_n\tilde{v}_i(X_{n+1/2}),
\end{aligned} \tag{OG+}$$

where $\gamma_n > \eta_n > 0$ are the respective learning rates for the extrapolation and update steps. Building on the intuition of Hsieh et al. (2022), when considering separated learning rates, the noise contaminating the gradient $\tilde{v}$ is effectively controlled ensuring that it is an order of magnitude smaller than the expected variation of payoffs. This stabilizes the learning dynamics in the setting of unconstrained monotone games with *first-order feedback*, whereby the algorithm enjoys last-iterate convergence of $X_{n+1/2}$ to a Nash equilibrium and attains an expected regret of $\tilde{\mathcal{O}}(\sqrt{n})$ under additive noise model and $\mathcal{O}(1)$ under multiplicative noise models.

## 3 SIMULTANEOUSLY PERTURBED OPTIMISTIC GRADIENT

We are now in a position to introduce *Simultaneously Perturbed Optimistic Gradient* (SPOG), an optimistic gradient algorithm for payoff-based learning in continuous games. To that end, we begin by coupling OG+ with SPSA+, defined by the update rule, for each $i \in \mathcal{N}$,

$$\begin{aligned}
Z_{n+1,i} &= X_{n,i} + \gamma_{n-1}V_{n,i}, \\
X_{n+1,i} &= X_{n,i} + \eta_n V_{n+1,i},
\end{aligned} \tag{OG+SPSA}$$

where the *adjusted* (joint) SPSA+ estimator $V_{n+1}$, given by,

$$V_{n+1,i} = \frac{D_i}{\delta_n}(u_i(\tilde{Z}_n) - u_i(\tilde{Z}_{n-1}))W_{n,i}, \tag{1}$$

and where $\tilde{Z}_n = Z_n + \delta_n W_n$, and $W_n$ is the joint perturbation for which each component $W_{n,i}$ is drawn independently of the other players and uniformly from the sphere $\mathbb{S}^{D_i}$.

In this scheme $Z_n$ takes the place of the extrapolation step $X_{n+1/2}$ in OG+ and is the action profile at which the pseudo-gradient $v(Z_n)$ is to be estimated. However, the variance introduced by the adjusted SPSA estimator 1 grows unbounded, which makes the iterates of the resultant algorithm impractical to control. This motivates a two-timescales approach (Borkar, 1997) where the extrapolation step $Z_n$ is updated with a larger learning parameter, effectively averaging across many gradient estimates and thereby controlling the variance of the SPSA estimator 1. The resulting update rule is:

$$\begin{aligned}
Z_{n+1,i} &= Z_{n,i} + \alpha_n(X_{n,i} + \gamma V_{n+1,i} - Z_{n,i}), \\
X_{n+1,i} &= X_{n,i} + \beta_n V_{n+1,i},
\end{aligned} \tag{2}$$

where $V_{n+1}$ is the *adjusted* (joint) SPSA estimator 1, $\gamma > 0$ is a fixed extrapolation parameter and $\alpha_n, \beta_n > 0$ with $\beta_n = o(\alpha_n)$ as $n \to \infty$ are the respective learning rates for the fast and slow

timescales. Owing to asymptotic difference in learning rates, we will refer to $X_n$ as the *slow iterate*, $Z_n$ as the *fast iterate*, which may be thought as a kind of 'time-average'. The *realized action* is $\tilde{Z}_n$.

Following the intuition of Borkar (1997), if the fast-process $Z_n$ converges for any fixed value of $X_n$ to a unique limit point, then we can analyze the algorithm as though the fast-process is, at each stage, fully calibrated to the current value of the slow process. To make this formal, consider the ordinary differential equation corresponding to the fast-process $Z_n$ as though the slow component is static at $X_n = x \in \mathcal{X}$, i.e.

$$\dot{z}(t) = x + \gamma v(z(t)) - z(t). \tag{ODE}$$

The parameter $\gamma > 0$ is subsequently tuned so that this ODE has a unique fixed point.

Unfortunately, 2 does not converge. The remaining challenge is that on unconstrained domains, concave functions are, in general, unbounded and not globally Lipschitz. We circumvent this by projecting the iterates into slowly-expanding envelopes, thus introducing a *deterministic* bound on the size of the iterates at a given time.

With this in hand, we are ready to present *Simultaneously Perturbed Optimistic Gradient* (SPOG). Let $(X_n)_{n \geq 1}$ and $(Z_n)_{n \geq 1}$ be the sequence of iterates defined by the update rule, for each $i \in \mathcal{N}$,

$$
\begin{aligned}
Z_{n+1,i} &= \mathrm{Proj}_{3R_{n+1}\mathbb{B}^{D_i}}[Z_{n,i} + \alpha_n(X_{n,i} + \gamma V_{n+1,i} - Z_{n,i})], \\
X_{n+1,i} &= \mathrm{Proj}_{R_{n+1}\mathbb{B}^{D_i}}[X_{n,i} + \beta_n V_{n+1,i}],
\end{aligned}
\tag{SPOG}
$$

where the *adjusted* (joint) SPSA+ estimator $V_{n+1}$, given by 1. See Algorithm 1 for pseudocode.

In addition, we impose the following assumptions on the various parameter sequences introduced.

**Assumption 2.** The sequences $\alpha_n, \beta_n, \delta_n > 0$ are *decreasing*, $\frac{\beta_n}{\alpha_n}$ is decreasing and converges to 0 as $n \to \infty$ and $\frac{\delta_{n-1}}{\delta_n}$ is uniformly bounded; and $0 < \gamma < \min\{\frac{1}{2L}, \frac{1}{2G\sqrt{N}}\}$. In addition, $\alpha_n, \beta_n, \delta_n, R_n$ satisfy

$$\lim_{n \to \infty} \alpha_n = \lim_{n \to \infty} \beta_n = \lim_{n \to \infty} \delta_n = \lim_{n \to \infty} \frac{\alpha_{n-1}R_n}{\delta_n} = 0, \ \lim_{n \to \infty} R_n = +\infty, \tag{3a}$$

$$\sum_{n=1}^{\infty} \alpha_n = +\infty \ , \sum_{n=1}^{\infty} \beta_n = +\infty, \sum_{n=1}^{\infty} \alpha_n\beta_n R_n^2 < \infty, \ \sum_{n=1}^{\infty} \frac{\beta_n^3 R_n^2}{\alpha_n^2} < \infty, \ \sum_{n=1}^{\infty} \beta_n\delta_n < \infty, \tag{3b}$$

In addition, we will sometimes restrict our study to parameter sequences of the following form.

**Assumption 3.** There exists constants $0 < a, b, d \leq 1$ and $\alpha, \beta, \delta, R > 0$ such that

$$\alpha_n = \frac{\alpha}{n^a}, \ \ \beta_n = \frac{\beta}{n^b}, \ \ \delta_n = \frac{\delta}{n^d}(\log n)^2, \ \text{and} \ R_n = R\log n. \tag{4}$$

*Remark.* Under Assumption 3, Assumption 2 is equivalent to the constants $a, b, d$ satisfying the following inequalities: $0 < d \leq a < b < 1$, $a + b > 1$, $b + d > 1$ and $3b - 2a > 1$. We will show below that the *optimal* choice of $a, b, d$ relative to the derived convergence guarantees in $n$ is to set $a = d = \frac{2}{3}$ and $b = 1$ and is *game-independent*. In practice, SPOG should be initialized with these exponents (or close enough, as per the discussion following Lemma 3.1).

As such the only parameter that must be tuned to the underlying game is $\gamma$, which must be chosen to satisfy $0 < \gamma < \min\{\frac{1}{2L}, \frac{1}{2G\sqrt{N}}\}$. To circumvent this requirement, one might consider a variable $\gamma$ approach, whereby $\gamma = \gamma_n$ converges to 0 on a third, even slower timescale. We opted for simplicity and avoided this extra layer of complication in our presentation of SPOG.

The parameter $R_n$ is introduced in order to project the iterates into a slowly growing envelope which we combine with the regularity Assumption 1(iv) in order to control the variance of the *adjusted* SPSA estimator $V_{n+1}$. It is necessary for our analysis that we project fast- and slow-timescales into envelopes of different radii. The following estimate underpins much of our analysis.

**Lemma 3.1.** *Under Assumptions 1-2, there exists a (deterministic) constant $C > 0$ such that, for sufficiently large $n$,*

$$\|V_{n+1}\| \leq CR_n. \tag{5}$$

---

**Algorithm 1** SPOG (player indices suppressed)

---

**Require:** learning rates $\alpha_n, \beta_n > 0$, query radius $\delta_n > 0$, parameter $\gamma > 0$
1: Choose $X, Z \in \mathcal{X}$, set $\tilde{u} \leftarrow 0$
2: **for** each stage $n = 1, 2, \ldots$ **do**
3:     draw $W$ uniformly from $\mathbb{S}^d$
4:     play $\tilde{Z} \leftarrow Z + \delta_n W$                              {default: $\delta_n \propto 1/n^{2/3}$}
5:     set $\tilde{u}^- \leftarrow \tilde{u}$
6:     receive $\tilde{u} = u(\tilde{Z})$
7:     set $\tilde{v} \leftarrow (d/\delta_n)(\tilde{u} - \tilde{u}^-) \cdot W$
8:     update $Z \leftarrow \text{Proj}_{3R_{n+1}\mathbb{B}^d}[Z + \alpha_n(X + \gamma\tilde{v} - Z)]$      {default: $\alpha_n \propto 1/n^{2/3}$}
9:     update $X \leftarrow \text{Proj}_{R_{n+1}\mathbb{B}^d}[X + \beta_n\tilde{v}]$            {default: $\beta_n \propto 1/n$}
10: **end for**

---

*Remark.* Here $C > 0$ is any constant satisfying $C > 2(1 + 8DG\sqrt{N} \sup_{k\geq 1} \frac{\delta_{k-1}}{\delta_k})$. In the proof of Lemma 3.1, we show that the upper bound 5 activates for all $n \geq n_1$, where

$$n_1 = \sup_{k \geq 1} \left\{ \frac{\alpha_{k-1} R_k}{\delta_k} > \frac{\min\{1, 2\sqrt{N}/\gamma\}}{16DGN} \right\}. \tag{6}$$

If SPOG is run with $a = d = \frac{2}{3}, b = 1$ as per Assumption 3, we have $\frac{\alpha_{k-1} R_k}{\delta_k} = \mathcal{O}(\frac{1}{\log n})$, so the number of rounds until 5 binds may be exponential in $D, G, N$ and $\gamma$. This is an artifact of the logarithmic scaling factor in $\delta_n$, and it can be avoided by taking

$$\alpha_n = \frac{\alpha}{n^{2/3}}, \quad \beta_n = \frac{\beta}{n}, \quad \delta_n = \frac{\delta}{n^{2/3-\epsilon}}, \quad \text{and} \quad R_n = R \log n. \tag{7}$$

where $\epsilon > 0$ is an arbitrary small constant. In this case, equation 5 binds after $n_1 = \text{poly}(D, G, N, \gamma)$ iterations, at the cost of only a slight deterioration in the algorithm's convergence rate. We discuss this issue in more detail right after the statement of Theorem 4.2.

The proof of this Lemma and all following statements are detailed in the Appendix.

## 4 RESULTS AND ANALYSIS

### 4.1 STATEMENT OF MAIN RESULTS

Our first key result is that SPOG converges to a Nash equilibrium in all monotone games.

**Theorem 4.1.** *Suppose that Assumptions 1-2 hold. Let $(X_n, Z_n)_{n\geq 1}$ be generated by SPOG. Then $X_n$ converges to a (possibly random) Nash equilibrium $x^\star \in \mathcal{X}_\star$ almost surely.*

There are two main steps to proving Theorem 4.1. First, in Section 4.2 we estimate the rate of convergence of the fast iterate $Z_n$ to a perturbed Nash equilibrium characterized by the slow iterate. In Section 4.3, we leverage the convergence of the fast iterate in order to analyze the asymptotic convergence of the slow iterate $X_n$.

Under the additional assumption that the game is *strongly*-monotone, we obtain a rate of convergence for the sequence $X_n$ generated by SPOG to the game's (unique) Nash equilibrium:

**Theorem 4.2.** *Suppose that Assumptions 1-3 hold, $\mathcal{G}$ is $\mu$-strongly monotone for some $\mu > 0$ and that $\gamma < \frac{1}{4\mu}$. Let $x^\star \in \mathcal{X}^\star$ be the (unique) Nash equilibrium of the game. Let $(X_n, Z_n)_{n\geq 1}$ be generated by SPOG. Then*

$$\mathbb{E} \|X_n - x^\star\|^2 = \tilde{\mathcal{O}}\left(n^{-f}\right), \tag{8}$$

*where $f = \min\{d, a, 2b - 2a\} > 0$.*

*Remark.* Under Assumption 3, the exponent $f = \min\{d, a, 2b-2a\}$ is maximized when $a = \frac{2}{3}, b = 1, d = \frac{2}{3}$, yielding $f = 2/3$. Hence the best-possible rate guarantee in 8 is

$$\mathbb{E} \|X_n - x^\star\|^2 = \tilde{\mathcal{O}}\left(n^{-2/3}\right). \tag{9}$$

This tuning has been chosen to optimize the dependence of the derived rate in $n$, and is asymptotic, as per the convergence rate guarantees of Bravo et al. (2018); Drusvyatskiy et al. (2022); Tatarenko and Kamgarpour (2024a); Amortila et al. (2024); Dong et al. (2025), upon which it improves (even though the cited results concern bounded domains). In the non-asymptotic regime, the worst-case constants in the $\tilde{\mathcal{O}}(1/n^{2/3})$ guarantee of equation 9 are determined by the precise time at which the variance bound of Lemma 3.1 activates, and equation 6 shows that these constants may carry an exponential dependence on $D$, $G$, $N$, and $\gamma$. Whether this is cause of concern or not depends on the size of the game (as captured by the product $DGN$), and the horizon of play $n$: if $DGN$ is too large, it might be preferable to choose a more conservative tuning for the algorithm's parameters $a$, $b$ and $d$, as per equation 7 for some small $\epsilon > 0$ (e.g., $\epsilon = 1/60$). In this case, the relevant constants stemming from equation 6 would be $\mathrm{poly}(D, G, N, \gamma)$ and the induced convergence rate guarantee would be $\mathbb{E}\|X_n - x^\star\|^2 = \tilde{\mathcal{O}}\left(n^{-2/3+\epsilon}\right)$—e.g., $\tilde{\mathcal{O}}(1/n^{13/20})$ if $\epsilon = 1/60$.

Calibrating the "sweet spot" in this trade-off is heavily application-dependent, so it lies beyond the scope of our work. We only note that, even for $\epsilon > 0$, the rate guarantees of Theorem 4.2 exceed the best-known $\mathcal{O}(n^{-1/2})$ rates in the literature, either anytime (Drusvyatskiy et al., 2022) or asymptotic (Tatarenko and Kamgarpour, 2024a), and even though these best-known rates only concern *bounded* domains (where the algorithm's exploration radius is bounded by default).

### 4.2 FAST-TIMESCALE ANALYSIS

In this section we establish, asymptotically, that the *fast iterate* $Z_n$ of SPOG is calibrated to the fixed point $z^\star(X_n)$ of the fast timescale mean field ODE at the current value of the *slow iterate* $X_n$.

**Lemma 4.3.** *For each fixed $x \in \mathbb{R}^D$, ODE has a unique globally attracting equilibrium $z^\star(x)$. Furthermore $z^\star : \mathbb{R}^D \to \mathbb{R}^D$ is Lipschitz, with Lipschitz constant $L_z = \frac{1}{1-\gamma L}$, and satisfies*

$$z^\star(x) = x + \gamma v(z^\star(x)). \tag{10}$$

With this result, we derive the convergence rate of the quantity $D_n \coloneqq \frac{\beta_n}{\alpha_n} \|Z_n - z^\star(X_n)\|^2$.

**Proposition 4.4.** *Suppose that Assumptions 1-2 hold. Let $(X_n, Z_n)_{n\geq 1}$ be generated by SPOG, then $D_n \to 0$ a.s. and in expectation as $n \to \infty$. If, in addition, the parameter sequences satisfy Assumption 3 then, for all $0 < \epsilon < \min\{b + d - a, 3b - 3a\}$,*

$$\mathbb{E}\left[\frac{\beta_n}{\alpha_n} \|Z_n - z^\star(X_n)\|^2\right] = \mathcal{O}\left(n^{-e+\epsilon}\right), \tag{11}$$

*where $e = \min\{b + d - a, b, 3b - 3a\}$.*

*Sketch of Proof.* Following Lyapunov's method (Benaïm, 2006), we first obtain a descent inequality that provides theoretical insight into performance benefits obtained through learning rate separation:

$$\mathbb{E}_n D_{n+1} \leq [1 - (\frac{1}{2} + 2\gamma\mu)\alpha_n]D_n \tag{12a}$$

$$+ \frac{2\gamma\beta_n}{\delta_n} \|b_{n+1}\|^2 + 2[\gamma^2\alpha_n\beta_n + L_z^2 \frac{\beta_n^3}{\alpha_n^2}]\mathbb{E}_n \|V_{n+1}\|^2. \tag{12b}$$

The full statement and proof of this descent inequality is given in Lemma E.1. By tuning the learning parameter sequences $\alpha_n, \beta_n, \delta_n$ such that each of the error terms in 12 is controlled sufficiently, we can apply a stochastic approximation argument in order to establish the asymptotic convergence of $D_n$ to zero. Moreover, owing to a contractive coefficient in 12a, we may apply Chung's Lemma B.3 to yield the last-iterate rate of convergence for $D_n$. $\square$

### 4.3 SLOW-TIMESCALE ANALYSIS

In this section we establish the last-iterate convergence of the *slow iterate* $X_n$ to a Nash equilibrium $x^\star \in \mathcal{X}_\star$ of the underlying game. First, we obtain the convergence of $\|X_n - x^\star\|^2$ *for any* Nash equilibrium $x^\star \in \mathcal{X}_\star$ by a stochastic approximation argument. Our proof also yields a kind of *best-iterate*, or *stabilization guarantee* for the convergence of the *fast iterate* $Z_n$. Last-iterate convergence of $X_n$ to an element of $\mathcal{X}_\star$ then follows as a consequence of a compactness argument.

**Proposition 4.5.** *Suppose that Assumptions 1-3 hold. Let $x^\star \in \mathcal{X}^\star$. Let $(X_n, Z_n)_{n \geq 1}$ be generated by SPOG. Then $\|X_n - x^\star\|^2$ converges to a finite random variable almost surely, and enjoys the stabilization guarantee $\sum_{n=1}^{\infty} \beta_n (\|v(Z_n)\|^2 + \|v(z^\star(X_n))\|^2) < \infty$ a.s.*

*Sketch of Proof.* Following a similar method to the fast-timescale analysis of Section 4.2, we begin by obtaining a descent-inequality for the Euclidean distance $\|X_{n+1} - x^\star\|^2$:

$$\mathbb{E}_n \|X_{n+1} - x^\star\|^2 \leq (1 - \mu\beta_n + \beta_n\delta_n) \|X_n - x^\star\|^2 \tag{13a}$$

$$+ \frac{1}{\gamma} \left( \frac{2\gamma\mu}{1 - 4\mu\gamma} + \gamma^2 L^2 + 2 \right) \beta_n \|Z - z^\star(X_n)\|^2 \tag{13b}$$

$$+ \frac{\beta_n}{\delta_n} \|b_{n+1}\|^2 + \beta_n^2 \mathbb{E}_n \|V_{n+1}\|^2 \tag{13c}$$

$$- \frac{1}{2}\gamma\beta_n \|v(Z_n)\|^2 - \frac{1}{2}\gamma\beta_n \|v(z^\star(X_n))\|^2 . \tag{13d}$$

The full statement and proof of this descent inequality is given in Lemma E.3. Critical to our subsequent analysis is the control of the calibration error term for the fast iterate 13b. By isolating this term in the descent inequality 13 we can utilize the convergence rate for this term obtained in Proposition 4.4. Assumptions 2 on the learning rate parameters enable the fast iterate to become calibrated whilst also controlling the error terms in 13. $\qquad\square$

## 5 EXPERIMENTS

In this section, we illustrate the last-iterate performance of SPOG with a comparison to OG+ in two simple two-player games, each with unique Nash equilibrium $x^\star = (0,0)$. We compare the performance SPOG with that of optimistic algorithms OG+ with *first-order* oracle (*additive* noise), as well OG+SPSA to serve as a zeroth-order comparator to SPOG, despite it not having theoretical convergence guarantees. We will compare the *update-step* of OG(+) $x_n$ with the *slow-iterate* of SPOG $X_n$ since these quantities are subject to theoretical convergence results Hsieh et al. (2020).

### 5.1 STRONGLY MONOTONE EXAMPLE

We illustrate the rate results of Theorem 4.2 for strongly monotone games with the following:

$$u_1(x_1, x_2) = -\frac{x_1^2}{2} - x_1 x_2 \ , \ u_2(x_1, x_2) = -\frac{x_2^2}{2} + x_1 x_2, \ \ x_1, x_2 \in \mathbb{R}. \tag{14}$$

In Figure 2a, after an initial transient phase, the algorithms appear to enter an asymptotic phase marked by the asymptotic log-linearity of the norm-squared. Both zeroth-order algorithms SPOG & OG+SPSA appear to exhibit faster asymptotic convergence than the first-order algorithms OG(+).

In Figure 2b trajectories of SPOG and OG+SPSA are noisy compared to those of OG(+). Unlike the instances of OG+ which query a noisy first-order oracle, each instance of SPOG cannot observe the gradient directly, instead having to estimate the gradient from payoff observations alone via an SPSA gradient estimator. Subsequently, SPOG makes random perturbations, producing this noise.

### 5.2 MERELY MONOTONE EXAMPLE

We illustrate the convergence result of Theorem 4.1 with this example of a two-player zero-sum bilinear game (which is necessarily *merely monotone*):

$$u_1(x_1, x_2) = x_1 x_2 = -u_2(x_1, x_2), \ \ x_1, x_2 \in \mathbb{R}. \tag{15}$$

In Figures 2c-2d SPOG appears to exhibit asymptotic convergence than any of its comparators, including OG+ which enjoys first-order gradient feedback. In this merely monotone experiment OG (without learning rate separation) does not converge, with trajectories orbiting the equilibrium in Figure 2d, illustrating the utility of learning rate separation (Hsieh et al., 2020). Moreover, it is unclear from $10^6$ iterations whether OG+SPSA converges, or instead leads to cycles of play. As alluded to in Section 2.4, the SPSA estimator 1 in OG+SPSA introduces large variance that is impractical to control, potentially leading to (at best) slow-, or non-convergence.

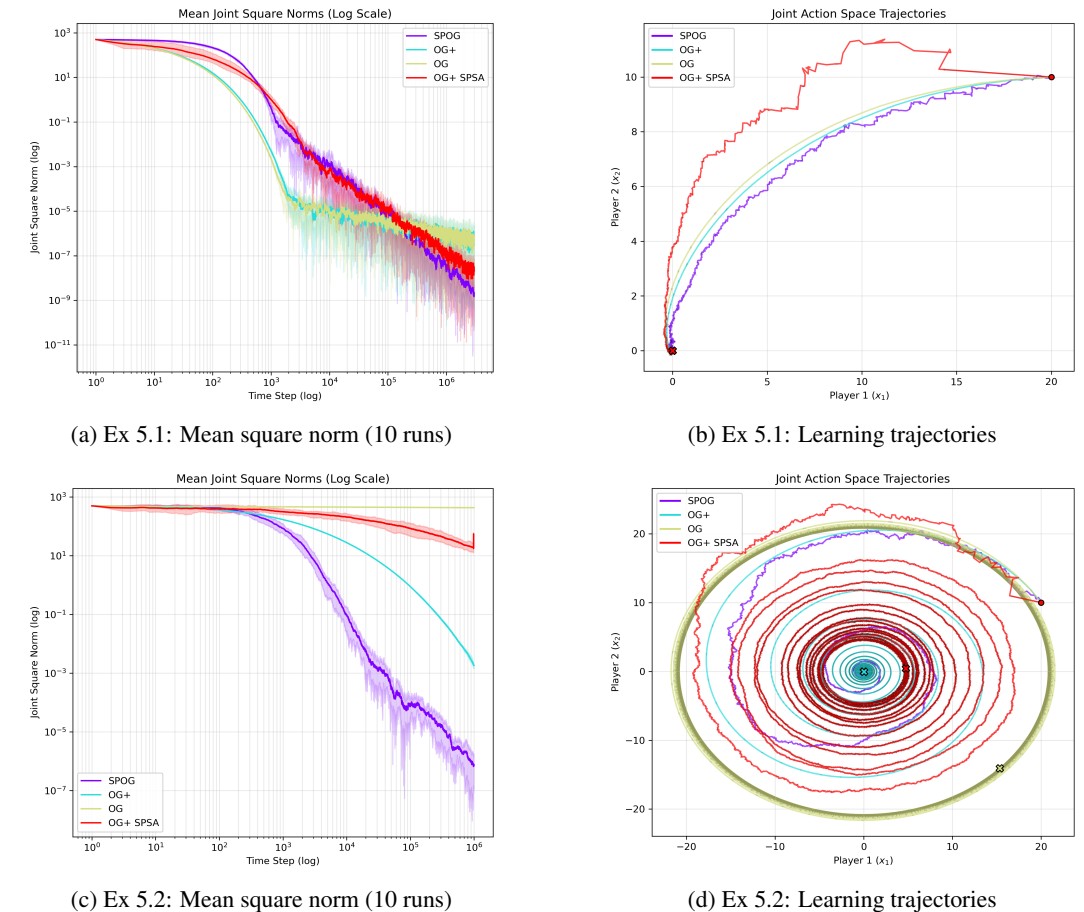

(a) Ex 5.1: Mean square norm (10 runs)

(b) Ex 5.1: Learning trajectories

(c) Ex 5.2: Mean square norm (10 runs)

(d) Ex 5.2: Learning trajectories

Figure 2: Distance to equilibrium in Example 5.1: SPOG vs OG+ vs OG vs OG+SPSA.

## 6 DISCUSSION

We developed a single-observation payoff-based algorithm whose iterates converge to a Nash equilibrium in all unconstrained monotone games and we established a *last-iterate* rate of convergence of $\tilde{\mathcal{O}}(1/n^{2/3})$ for the sequence of iterates in strongly monotone games. This rate exceeds the best known rate for a single-observation payoff-based algorithm in the constrained setting (Drusvyatskiy et al., 2022; Tatarenko and Kamgarpour, 2024a). We note that this exceeds the optimal lower complexity bound of $\Omega(n^{-1/2})$ for one-point zeroth-order algorithms (Shamir, 2013; Ba et al., 2025). We believe that this discrepancy is the result of using an adjusted SPSA gradient estimate, SPSA+, as opposed to using the traditional SPSA estimate which has been assumed in the literature. Our adjusted SPSA estimator reuses previous payoff observations, effectively making use of two payoff queries. Nonetheless, we argue that SPOG remains within the category of *"single-shot zeroth-order"* learning algorithms as as *only one payoff observation is made at each iteration.*

Our algorithm employs a *learning-rate separation* technique (Hsieh et al., 2022) which we view as an instance of two-timescales stochastic approximation (Borkar, 1997). This technique is particularly useful in the zeroth-order framework, where the variance of the pseudo-gradient estimate grows to be unbounded. In effect, by averaging across many gradient estimates on a fast timescale, our algorithm controls the variance. In the unconstrained setting, first-order algorithms such as Hsieh et al. (2022) require that the noise contaminating gradient feedback has finite variance, thereby avoiding this problem altogether. In the analysis of payoff-based algorithm in *constrained* games (Bravo et al., 2018; Tatarenko and Kamgarpour, 2024a) control of the variance relies on the compactness of the game's action spaces. With neither option being available in monotone games with *unconstrained* action spaces, we believe the learning-rate separation is critical to the convergence of the iterates.

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

## A  NOTATION

We summarize our notation in Table 2.

## B  STOCHASTIC APPROXIMATION THEORY

We present a stochastic approximation theorem attributed to Robbins and Siegmund.

**Theorem B.1** (Robbins (1975))**.** *Let $(\Omega, \mathcal{F}, \mathbb{P})$ be a probability space and let $\mathcal{F}_1 \subset \mathcal{F}_2 \subset \ldots$ be a sequence of sub-$\sigma$-fields of $\mathcal{F}$. Let $U_n, \beta, \xi_n, \zeta_n,\ n \in \mathbb{N}$ be non-negative $\mathcal{F}_n$-measurable random variables satisfying $\mathbb{E}[U_1] < \infty$ and*

$$\mathbb{E}[U_{n+1}|\mathcal{F}_n] \leq (1 + \beta_n)U_n + \xi_n - \zeta_n,\ n = 1, 2, \ldots$$

*Suppose that $\sum_{n=1}^{\infty} \mathbb{E}[\beta_n] < +\infty$ and $\sum_{n=1}^{\infty} \mathbb{E}[\xi_n] < +\infty$. Then, $U_n$ converges a.s. to a finite random variable and $\sum_{n=1}^{\infty} \zeta_n < \infty$ a.s.*

Another theorem of stochastic approximation is the following, and is proven in Lemma 10 (page 49) of Polyak (1987)

Table 2: Notations

| Symbol | Description |
|--------|-------------|
| $N$ | Number of players |
| $\mathcal{N}$ | Set of player indices $\mathcal{N} = \{1, \ldots, N\}$ |
| $\mathcal{X}_i$ | Strategy space for player $i$, $\mathcal{X}_i = \mathbb{R}^{D_i}$ |
| $D_i$ | Dimension of player $i$'s action space |
| $\mathcal{X}$ | Joint strategy space: $\mathcal{X} = \prod_{i=1}^{N} \mathcal{X}_i$ |
| $d$ | Dimension of joint action space |
| $u_i$ | Payoff function for player $i$ |
| $v_i$ | Individual payoff gradient for player $i$ |
| $x^\star$ | Nash equilibrium |
| $\mathcal{X}^\star$ | Set of Nash equilibria |
| $L$ | Lipschitz constant of $(v_i)_{i \in \mathcal{N}}$ |
| $\mu$ | Monotonicity constant of $(v_i)_{i \in \mathcal{N}}$ |
| $G$ | Smoothness constant of full gradient $(\nabla u_i)_{i \in \mathcal{N}}$ |
| $r\mathbb{B}^{D_i}$ | $D_i$-dimensional ball of radius $r$, $r\mathbb{B}^{D_i} = \{p \in \mathbb{R}^{D_i} : \|p\| \leq r\}$ |
| $\mathbb{S}^{D_i}$ | $D_i$-dimensional unit sphere $\mathbb{S}^{D_i} = \{p \in \mathbb{R}^{D_i} : \|p\| = 1\}$ |
| $Z_n$ | Fast timescale iterate at time $n$ |
| $X_n$ | Slow timescale iterate at time $n$ |
| $\tilde{Z}_n$ | Realized joint action profile at time $n$ |
| $V_{n+1}$ | Adjusted joint SPSA estimator |
| $\delta_n$ | SPSA perturbation parameter |
| $\alpha_n$ | Fast-timescale learning rate |
| $\beta_n$ | Slow-timescale learning rate |
| $R_n$ | Radius feasible envelope $R_n = R \log n$ |
| $\gamma$ | Contraction parameter for fast timescale update |
| $\mathcal{F}_n$ | $\sigma$-algebra generated by history of play $X_1, Z_1, W_1, \ldots, X_n, Z_n, W_n$ |
| $\mathbb{E}_n$ | Expectation with respect to $\mathcal{F}_n$, $\mathbb{E}_n[\cdot] = \mathbb{E}[\cdot|\mathcal{F}_n]$ |
| $z^\star(x)$ | Fast-timescale ODE fixed point |
| $L_z$ | Lipschitz constant of $z^\star$, $L_z = \frac{1}{1-\gamma L}$ |

**Theorem B.2.** *Let $(\Omega, \mathcal{F}, \mathbb{P})$ be a probability space and let $\mathcal{F}_1 \subset \mathcal{F}_2 \subset \ldots$ be a sequence of sub-$\sigma$-fields of $\mathcal{F}$. Let $U_n, \beta, \xi_n, n \in \mathbb{N}$ be non-negative $\mathcal{F}_n$-measurable random variables satisfying $\mathbb{E}[U_1] < \infty$ and*

$$\mathbb{E}[U_{n+1}|\mathcal{F}_n] \leq (1 - \beta_n)U_n + \xi_n, \ n = 1, 2, \ldots$$

*Suppose that $\sum_{n=1}^{\infty} \mathbb{E}[\beta_n] = +\infty$, $\sum_{n=1}^{\infty} \mathbb{E}[\xi_n] < +\infty$, $0 < \beta_n < 1$ and $\xi_n \geq 0$. Then, $U_n \to 0$ a.s. and $\mathbb{E}[U_n] \to 0$ as $n \to \infty$.*

We will also make use of the following Lemma on numerical sequences when it comes to obtaining rates of convergence. This is often referred to and attributed to in the literature as Chung's Lemma (Chung, 1954).

**Lemma B.3.** *[Chung's Lemma (Chung, 1954)] Let $(a_n)_{n \in \mathbb{N}}$ be a non-negative sequence satisfying*

$$a_{n+1} \leq \left(1 - \frac{P}{n^p}\right)a_n + \frac{Q}{n^{p+q}},$$

*where $0 < p \leq 1, q > 0$ and $P, Q > 0$ and assuming in addition that $P > q$ if $p = 1$. Then we have that*

$$a_n \leq \frac{Q}{R}\frac{1}{n^q} + o\left(\frac{1}{n^q}\right),$$

*with*

$$R = \begin{cases} P & \text{if } p < 1, \\ P - q & \text{if } p = 1. \end{cases}$$

## C  Properties of the Gradient Estimate $V_{n+1}$

### C.1  Proof of Lemma 3.1

*Proof of Lemma 3.1.* Let $(X_n, Z_n)_{n \geq 1}$ be generated by SPOG.

Fix $i \in \mathcal{N}$. Owing to the Mean Value Theorem, there exists a $t_n \in [0,1]$ such that

$$u_i(\tilde{Z}_n) - u_i(\tilde{Z}_{n-1}) = \langle \nabla u_i(t\tilde{Z}_n + (1-t)\tilde{Z}_{n-1}), \tilde{Z}_n - \tilde{Z}_{n-1} \rangle. \tag{16}$$

Applying the Cauchy-Schwartz inequality, and Assumption 1(iv) on the gradient $\nabla u_i$, we arrive at the following.

$$\begin{aligned}
\|V_{n+1,i}\| &= \frac{D_i}{\delta_n} |u_i(\tilde{Z}_n) - u_i(\tilde{Z}_{n-1})| \\
&= \frac{D_i}{\delta_n} |\langle \nabla u_i(t_n\tilde{Z}_n + (1-t_n)\tilde{Z}_{n-1}), \tilde{Z}_n - \tilde{Z}_{n-1} \rangle| \\
&\leq \frac{D_i}{\delta_n} \left\| \nabla u_i(t_n\tilde{Z}_n + (1-t_n)\tilde{Z}_{n-1}) \right\| \left\| \tilde{Z}_n - \tilde{Z}_{n-1} \right\| \\
&\leq \frac{D_i G}{\delta_n} \left( 1 + \left\| t_n\tilde{Z}_n + (1-t_n)\tilde{Z}_{n-1} \right\| \right) \left\| \tilde{Z}_n - \tilde{Z}_{n-1} \right\|. \tag{17}
\end{aligned}$$

Applying the triangle inequality to each of the norms, we have that equation 17 implies

$$\|V_{n+1,i}\| \leq \frac{D_i G}{\delta_n}[1 + t_n\delta_n + (1-t_n)\delta_{n-1} + t_n\|Z_n\| + (1-t_n)\|Z_{n-1}\|][\delta_n + \delta_{n-1} + \|Z_n - Z_{n-1}\|] \tag{18}$$

As a result of the projections in SPOG, we have that $\|Z_{n,i}\| \leq 3R_n$, whence $\|Z_n\| \leq 3\sqrt{N}R_n$. Similarly, $\|Z_{n-1}\| \leq 3\sqrt{N}R_{n-1}$. Following Assumption 2, $\delta_n$ is a decreasing sequence, and $R_n$ is an increasing sequence, we obtain the following inequality from equation 18:

$$\|V_{n+1,i}\| \leq \frac{D_i G}{\delta_n}(1 + \delta_{n-1} + 3\sqrt{N}R_n)(2\delta_{n-1} + \|Z_n - Z_{n-1}\|) \tag{19}$$

Finally, by first setting $\mathcal{B}_m := 3R_m\mathbb{B}^{D_i}$ for each $m \geq 1$, remark that $Z_n = \text{Proj}_{\prod_i 3R_n\mathbb{B}^{D_i}} Z_n^o$, and $Z_{n-1} \in \prod_i 3R_{n-1}\mathbb{B}^{D_i} \subseteq \prod_i 3R_n\mathbb{B}^{D_i} = \mathcal{B}_n$. By the non-expansiveness of the projection operator, we have that

$$\begin{aligned}
\|Z_n - Z_{n-1}\| &= \left\| \text{Proj}_{\mathcal{B}_n} Z_n^o - \text{Proj}_{\mathcal{B}_n} Z_{n-1} \right\| \\
&\leq \|Z_n^o - Z_{n-1}\| \\
&= \alpha_{n-1}\|X_{n-1} - Z_{n-1} + \gamma V_n\| \\
&\leq \alpha_{n-1}\|X_{n-1}\| + \alpha_{n-1}\|Z_{n-1}\| + \alpha_{n-1}\gamma\|V_n\| \\
&\leq 4\alpha_{n-1}\sqrt{N}R_n + \gamma\alpha_{n-1}\|V_n\| \tag{20}
\end{aligned}$$

where the final inequality again follows from the projections in SPOG. Combined with equation 19, we arrive at the following estimate:

$$\|V_{n+1,i}\| \leq \frac{D_i G}{\delta_n}(1 + \delta_{n-1} + 3\sqrt{N}R_n)(2\delta_{n-1} + 4\alpha_{n-1}\sqrt{N}R_n + \gamma\alpha_{n-1}\|V_n\|)$$

Since $\sqrt{D_1^2 + \cdots + D_n^2} \leq D_1 + \cdots + D_n = D$, and $i \in \mathcal{N}$ was chosen arbitrarily, we obtain the following inequality for $\|V_{n+1}\|$,

$$\|V_{n+1}\| \leq \frac{DG}{\delta_n}(1 + \delta_{n-1} + 3\sqrt{N}R_n)(2\delta_{n-1} + 4\alpha_{n-1}\sqrt{N}R_n) + \gamma\frac{DG}{\delta_n}(1 + \delta_{n-1} + 3\sqrt{N}R_n)\alpha_{n-1}\|V_n\|. \tag{21}$$

As a result of Assumption 2, there exists a finite (deterministic) $n_0$ such that for all $n \geq n_0$, $1 + \delta_{n-1} < \sqrt{N}R_n$. Substituting this into equation 21 and setting $M := 4DG\sqrt{N}$ we arrive at the following inequality for all $n \geq n_0$,

$$\|V_{n+1}\| \leq \frac{M}{\delta_n}R_n(2\delta_{n-1} + 4\alpha_{n-1}\sqrt{N}R_n) + \frac{\gamma M}{\delta_n}R_n\alpha_{n-1}\|V_n\|. \tag{22}$$

Following Assumption 2, there exists a uniform bound $\Delta > 0$ such that for all $n$, $\frac{\delta_{n-1}}{\delta_n} \leq \Delta$. Hence for all $n \geq n_0$,

$$\|V_{n+1}\| \leq 2M\Delta R_n + 4M\sqrt{N}\frac{\alpha_{n-1}R_n^2}{\delta_n} + \gamma M\frac{\alpha_{n-1}R_n}{\delta_n}\|V_n\|. \tag{23}$$

Owing to the Assumption 2 that $\frac{\alpha_{n-1}R_n}{\delta_n} \to 0$ as $n \to \infty$, we have that there exists a finite (deterministic) $n_1 \geq n_0$ such that for all $n \geq n_1$, $\frac{\alpha_{n-1}R_n}{\delta_n} \leq \min\{\frac{1}{4M\sqrt{N}}, \frac{1}{2\gamma M}\}$. Hence, for all $n \geq n_1$,

$$\|V_{n+1}\| \leq (2M\Delta + 1)R_n + \frac{1}{2}\|V_n\|. \tag{24}$$

As a result of equation 24 and the increasing property of $R_n$, we have that for all $n \geq n_1$, and all $C > 2(2M\Delta + 1)$,

$$\|V_n\| \leq CR_{n-1} \implies \|V_{n+1}\| \leq (2M\Delta + 1 + \frac{1}{2}C)R_n \leq CR_n. \tag{25}$$

In particular, since $n_1$ is deterministic and finite and

$$\|V_{n_1}\| = \sqrt{\frac{D_i^2}{\delta_{n_1}^2}|u_i(\tilde{Z}_{n_1}) - u_i(\tilde{Z}_{n_1-1})|^2} \tag{26}$$

is bounded by a deterministic constant, owing to the fact that each $u_i$ is continuous and $\tilde{Z}_{n_1-1}, \tilde{Z}_{n_1}, \in \prod_i(3R_{n_1} + \delta_{n_1})\mathbb{B}^{D_i}$. Hence there exists a $C > 2(2M\Delta + 1)$ such that $\|V_{n_1}\| \leq CR_{n_1}$ and the result follows by induction. $\qquad\square$

# D PROPERTIES OF THE FIXED POINT $z^\star$

## D.1 PROOF OF LEMMA 4.3

*Proof of Lemma 4.3.* Fix $x \in \mathbb{R}^D$. We begin by remarking that the function $f_x(z) = x + \gamma v(z)$ satisfies

$$\|f_x(z_1) - f_x(z_2)\| = \gamma\|v(z_1) - v(z_2)\| \leq \gamma L\|z_1 - z_2\|.$$

Since $\gamma L < 1$, the Banach fixed point theorem (Agarwal et al., 2018) implies the existence of a unique fixed point $z^\star(x)$ satisfying $f_x(z^\star(x)) = z^\star(x)$. Such a fixed point is an equilibrium of the ODE by construction.

We define a Lyapunov function $\Lambda$ for the fixed $x$ ODE as follows $\Lambda(t) = \frac{1}{2}\|z(t) - z^\star(x)\|$. We have that

$$\begin{aligned}
\frac{d\Lambda(t)}{dt} &= \langle \dot{z}(t), z(t) - z^\star(x)\rangle \\
&= \langle x + \gamma v(z(t)) - z(t), z(t) - z^\star(x)\rangle \\
&= \gamma\langle v(z(t)) - v(z^\star(x)), z(t) - z^\star(x)\rangle - \langle z(t) - z^\star(x), z(t) - z^\star(x)\rangle \\
&\leq -\|z(t) - z^\star(x)\|^2,
\end{aligned}$$

where the final inequality follows from the monotonicity of $v$. This shows that the Lyapunov function $\Lambda$ is strict and so the equilibrium $z^\star(x)$ is globally stable.

Finally, we note that

$$z^\star(x) = x + \gamma v(z^\star(x))$$

and so, for any $x_1, x_2 \in \mathbb{R}^D$,

$$\begin{aligned}
\|z^\star(x_1) - z^\star(x_2)\| &= \|(x_1 - x_2) + \gamma(v(z^\star(x_1)) - v(z^\star(x_2)))\| \\
&\leq \|x_1 - x_2\| + \gamma\|v(z^\star(x_1)) - v(z^\star(x_1))\| \\
&\leq \|x_1 - x_2\| + \gamma L\|z^\star(x_1) - z^\star(x_2)\|,
\end{aligned}$$

where the last line follows from the Lipschitz continuity of $v$. Rearranging, we obtain the Lipschitz condition for $z^\star$,

$$\|z^\star(x_1) - z^\star(x_2)\| \leq \frac{1}{1 - \gamma L}\|x_1 - x_2\|.$$

$\square$

## D.2 FIXED POINT INCLUSION

The following Lemma concerns the image of the region $\prod_i R\mathbb{B}^{D_i}$ under $z^\star$.

**Lemma D.1.** *Suppose that Assumptions 1-2 hold. For all $R > 1$ the following inclusion holds*

$$z^\star(\prod_{i\in\mathcal{N}} R\mathbb{B}^{D_i}) \subseteq \prod_{i\in\mathcal{N}} 3R\mathbb{B}^{D_i}.$$

*Remark.* This Lemma is the reason that in SPOG project $X_{n,i}^o$ into $R_n\mathbb{B}^{D_i}$ and $Z_{n,i}^o$ into $3R_n\mathbb{B}^{D_i}$. In our convergence analysis of SPOG we will consider the distance $\|Z_n - z^\star(X_n)\|^2$, which, as a result of this lemma, is comparing two points in $\prod_{i\in\mathcal{N}} 3R_n\mathbb{B}^{D_i}$. Hence we may apply the non-expansiveness of the projection operator to extract $X_n^o$.

*Proof.* Let $x \in \prod_i R\mathbb{B}^{D_i}$. We have that $v_i(z) = \nabla_i u_i(z)$ for all $z \in \mathbb{R}^{D_i}$ and all $i \in \mathcal{N}$. As a result of Assumption 1(iv) we have that

$$
\begin{aligned}
\|v(z^\star(x))\| &= \sqrt{\sum_{i=1}^N \|\nabla_i u_i(z^\star(x))\|_{\mathbb{R}^{D_i}}^2} \\
&\leq \sqrt{\sum_{i=1}^N \|\nabla u_i(z^\star(x))\|_{\mathbb{R}^d}^2} \\
&\leq \sqrt{\sum_{i=1}^N G^2(1 + \|z^\star(x)\|)^2} \\
&= G\sqrt{N}(1 + \|z^\star(x)\|)
\end{aligned}
\tag{27}
$$

Since $z^\star(x) = x + \gamma v(z^\star(x))$, we may apply the triangle inequality to obtain that

$$
\begin{aligned}
\|z^\star(x)\| &\leq \|x\| + \gamma \|v(z^\star(x))\| \\
&\leq \|x\| + \gamma G\sqrt{N}(1 + \|z^\star(x)\|).
\end{aligned}
\tag{28}
$$

where the final inequality follows from equation 27. Rearranging the inequality equation 28, we arrive at the following

$$\|z^\star(x)\| \leq \frac{\gamma G\sqrt{N}}{1 - \gamma G\sqrt{N}} + \frac{1}{1 - \gamma G\sqrt{N}} \|x\|.$$

As a consequence of Assumption 2, we have that $\gamma G\sqrt{N} \leq \frac{1}{2}$. Since the function $\psi \mapsto \frac{1}{1-\psi}$ is increasing on the interval $(0, \frac{1}{2}]$, we have that

$$\|z^\star(x)\| \leq \frac{\gamma G\sqrt{N}}{1 - \gamma G\sqrt{N}} + \frac{1}{1 - \gamma G\sqrt{N}} \|x\| \leq 1 + 2\|x\|.$$

Since we have assumed that $1 \leq R$ and $\|x\| \leq R$, we conclude $\|z^\star(x)\| \leq 3R$. Hence we have that $z^\star(\prod_{i\in\mathcal{N}} R\mathbb{B}^{D_i}) \subseteq 3R\mathbb{B}^d \subseteq \prod_{i\in\mathcal{N}} 3R\mathbb{B}^{D_i}$. □

## D.3 FIXED-POINT VARIATIONAL INEQUALITY

The following Lemma is a property of the underlying game and the fixed-point function $z^\star$. It will feature in our analysis.

**Lemma D.2.** *Suppose that Assumption 1 holds. Suppose that $\gamma L < 1$ and that $x^\star \in \mathcal{X}^\star$ is a Nash equilibrium of the underlying game. If the game is strongly monotone, suppose in addition that $\gamma\mu < \frac{1}{4}$. Then for all $z, x \in \mathcal{X}$ the following holds:*

$$\langle v(z), x - x^\star \rangle \leq -\frac{\mu}{2}\|x - x^\star\|^2 + \frac{1}{2\gamma}\left(\frac{2\gamma\mu}{1 - 4\mu\gamma} + \gamma^2 L^2 + 2\right)\|z - z^\star(x)\|^2 - \frac{1}{4}\gamma\|v(z)\|^2 - \frac{1}{4}\gamma\|v(z^\star(x))\|^2$$

$$\tag{29}$$

*Proof.* Fix $z, x \in \mathcal{X}$. Writing $x - x^\star = (x - z) + (z - x^\star)$, we have that

$$\langle v(z), x - x^\star \rangle = \langle v(z), x - z \rangle + \langle v(z), z - x^\star \rangle \tag{30}$$

We handle each of these terms separately.

First, we consider the term $\langle v(z), z - x^\star \rangle$. Since $x^\star \in \mathcal{X}^\star$ is a Nash equilibrium, $v(x^\star) = 0$. This, combined with the $\mu$-monotonicity of the underlying game implies that

$$\langle v(z), z - x^\star \rangle = \langle v(z) - v(x^\star), z - x^\star \rangle \leq -\mu \|z - x^\star\|^2. \tag{31}$$

Again, writing $z - x^\star = (z - x) + (x - x^\star)$, we expand the norm as follows:

$$-\mu \|z - x^\star\|^2 = -\mu \|x - x^\star\| - \mu \|z - x\|^2 - 2\mu\langle z - x, x - x^\star \rangle \tag{32}$$

An application of Young's inequality for products implies that, for any $\epsilon > 0$

$$-2\mu\langle z - x, x - x^\star \rangle \leq \frac{1}{\epsilon}\mu \|z - x\|^2 + \epsilon\mu \|x - x^\star\|^2. \tag{33}$$

Setting $\epsilon = \frac{1}{2}$ in equation 33, we obtain that

$$-2\mu\langle z - x, x - x^\star \rangle \leq 2\mu \|z - x\|^2 + \frac{1}{2}\mu \|x - x^\star\|^2. \tag{34}$$

Applying equation 34 to the inner product in equation 32 and using the result in equation 31, we arrive at the following inequality

$$\langle v(z), z - x^\star \rangle \leq -\frac{\mu}{2} \|x - x^\star\| + \mu \|z - x\|^2. \tag{35}$$

For the term $\mu \|z - x\|^2$, we express apply the definition of the fixed point $z^\star$ to write $x = z^\star(x) - \gamma v(z^\star(x))$. Subsequent application of Young's inequality for products yields, for any $\theta > 0$,

$$\mu \|z - x\|^2 = \|z - z^\star(x) + \gamma v(z^\star(x))\|^2 \leq (1 + \theta)\mu \|z - z^\star(x)\|^2 + (1 + \theta^{-1})\gamma^2\mu \|v(z^\star(x))\|^2. \tag{36}$$

Setting $\theta = \frac{4\gamma\mu}{1 - 4\gamma\mu}$, we observe that $1 + \theta^{-1} = \frac{1}{4\gamma\mu}$ and $1 + \theta = \frac{1}{1 - 4\gamma\mu}$. Hence equation 36 becomes

$$\mu \|z - x\|^2 = \|z - z^\star(x) - \gamma v(z^\star(x))\|^2 \leq \frac{\mu}{1 - 4\gamma\mu} \|z - z^\star(x)\|^2 + \frac{1}{4}\gamma \|v(z^\star(x))\|^2. \tag{37}$$

Applying equation 37 to the right hand side of equation 35, we arrive at the following inequality for the Nash term of equation 30

$$\langle v(z), z - x^\star \rangle \leq -\frac{\mu}{2} \|x - x^\star\| + \frac{\mu}{1 - 4\gamma\mu} \|z - z^\star(x)\|^2 + \frac{1}{4}\gamma \|v(z^\star(x))\|^2. \tag{38}$$

To handle the remaining term in equation 30, $\langle v(z), x - z \rangle$, we again write $x = z^\star(x) - \gamma v(z^\star(x))$. Then

$$\langle v(z), x - z \rangle = \langle v(z), z^\star(x) - z - \gamma v(z^\star(x)) \rangle = \langle v(z), z^\star(x) - z \rangle - \gamma\langle v(z), v(z^\star(x)) \rangle. \tag{39}$$

By simply expanding the norm, we see that

$$-\gamma\langle v(z), v(z^\star(x)) \rangle = \frac{1}{2}\gamma \|v(z^\star(x)) - v(z)\|^2 - \frac{1}{2}\gamma \|v(z)\|^2 - \frac{1}{2}\gamma \|v(z^\star(x))\|^2$$

$$\leq \frac{1}{2}\gamma L^2 \|z^\star(x) - x\|^2 - \frac{1}{2}\gamma \|v(z)\|^2 - \frac{1}{2}\gamma \|v(z^\star(x))\|^2, \tag{40}$$

where the inequality is a result of the Lipschitz continuity of $v$. To handle the remaining term of equation 39, $\langle v(z), z^\star(x) - z \rangle$, we apply Young's inequality for products, obtaining,

$$\langle v(z), z^\star(x) - z \rangle \leq \frac{1}{2} \cdot \frac{1}{2}\gamma \|v(z)\|^2 + \frac{1}{2}\frac{1}{\left(\frac{1}{2}\gamma\right)} \|z - z^\star(x)\|^2 = \frac{1}{4}\gamma \|v(z)\|^2 + \frac{1}{\gamma} \|z - z^\star(x))\|^2. \tag{41}$$

Applying equation 40 and equation 41 to equation 39, we arrive at the following inequality

$$\langle v(z), x - z \rangle \leq \left(\frac{1}{2}\gamma L^2 + \frac{1}{\gamma}\right) \|z - z^\star(x)\|^2 - \frac{1}{4}\gamma \|v(z)\|^2 - \frac{1}{2}\gamma \|v(z^\star(x))\|^2. \tag{42}$$

Applying both equation 38 and equation 42 to equation 30 yields the desired result. $\qquad\square$

## E    DERIVATION OF DESCENT INEQUALITIES

**Lemma E.1** (Fast-Descent Inequality). *Under Assumptions 1-2,*

$$\mathbb{E}_n \left\| Z_{n+1} - z^\star(X_{n+1}) \right\|^2 \leq \left( 1 - (1 + 2\gamma\mu - \gamma\delta_n)\alpha_n - (1 - 2\gamma^2 L^2 - \gamma\delta_n)\alpha_n^2 + (1 + 2\gamma\mu + 2\gamma^2 L^2)\alpha_n^3 \right) \left\| Z_n - z^\star(X_n) \right\|^2$$
$$+ (1 + \alpha_n)(\gamma^2\alpha_n^2 + \frac{\gamma\alpha_n}{\delta_n}) \left\| b_{n+1} \right\|^2$$
$$+ \left( \gamma^2\alpha_n^2(1 + \alpha_n) + L_z^2\beta_n^2(1 + \frac{1}{\alpha_n}) \right) \mathbb{E}_n \left\| V_{n+1} \right\|^2.$$

*Proof.* As a consequence of Young's inequality for products, we have that for any $\theta_n > 0$,

$$\left\| Z_{n+1} - z^\star(X_{n+1}) \right\|^2 = \left\| Z_{n+1} - z^\star(X_n) + z^\star(X_n) - z^\star(X_{n+1}) \right\|^2$$
$$\leq (1 + \theta_n) \left\| Z_{n+1} - z^\star(X_n) \right\|^2 + (1 + \frac{1}{\theta_n}) \left\| z^\star(X_{n+1}) - z^\star(X_n) \right\|^2. \tag{44}$$

We handle each of the terms in equation 44 separately. First, we remark that by the Lipchitz continuity of the fixed point $z^\star$,

$$\left\| z^\star(X_{n+1}) - z^\star(X_n) \right\|^2 \leq L_z^2 \left\| X_{n+1} - X_n \right\|^2$$
$$= L_z^2 \left\| \text{Proj}_{R_{n+1}\mathbb{B}^d} X_{n+1}^o - X_n \right\|^2$$
$$= L_z^2 \left\| \text{Proj}_{R_{n+1}\mathbb{B}^d} X_{n+1}^o - \text{Proj}_{R_{n+1}\mathbb{B}^d} X_n \right\|^2$$
$$\leq L_z^2 \left\| X_{n+1}^o - X_n \right\|^2 = L_z^2\beta_n^2 \left\| V_{n+1} \right\|^2, \tag{45}$$

where we have used the fact that $X_n \in R_n\mathbb{B}^d \subseteq R_{n+1}\mathbb{B}^d$, and that the projection operator is non-expansive.

To handle the first term in equation 44, we first remark that, as a result of Lemma D.1, $z^\star(X_n) \in \prod_i 3R_n\mathbb{B}^{D_i} \subseteq \prod_i 3R_{n+1}\mathbb{B}^{D_i}$. Again applying the non-expansiveness of the projection operator, we have that

$$\left\| Z_{n+1} - z^\star(X_n) \right\|^2 = \left\| \text{Proj}_{3R_{n+1}\mathbb{B}^d} Z_{n+1}^o - \text{Proj}_{3R_{n+1}\mathbb{B}^d} z^\star(X_n) \right\|^2 \leq \left\| Z_{n+1}^o - z^\star(X_n) \right\|^2. \tag{46}$$

Next, we observe that, by rearranging the fixed point formula, $X_n = z^\star(X_n) - \gamma v(z^\star(X_n))$. With this in hand, we obtain the following:

$$Z_{n+1}^o - z^\star(X_n) = Z_n + \alpha_n(X_n - Z_n + \gamma V_{n+1}) - z^\star(X_n)$$
$$= Z_n - z^\star(X_n) + \alpha_n[z^\star(X_n) - Z_n - \gamma v(z^\star(X_n)) + \gamma V_{n+1}]$$
$$= (1 - \alpha_n)(Z_n - z^\star(X_n)) - \gamma\alpha_n v(z^\star(X_n)) + \gamma\alpha_n V_{n+1}. \tag{47}$$

Substituting equation 47 into equation 46 and expanding the norm, we have that

$$\left\| Z_{n+1}^o - z^\star(X_n) \right\|^2 \leq \left\| (1 - \alpha_n)(Z_n - z^\star(X_n)) - \gamma\alpha_n v(z^\star(X_n)) \right\|^2 \tag{48a}$$
$$+ 2\gamma\alpha_n \langle (1 - \alpha_n)(Z_n - z^\star(X_n)) - \gamma\alpha_n v(z^\star(X_n)), V_{n+1} \rangle \tag{48b}$$
$$+ \gamma^2\alpha_n^2 \left\| V_{n+1} \right\|^2. \tag{48c}$$

We first expand the term equation 48a

$$\left\| (1 - \alpha_n)(Z_n - z^\star(X_n)) - \gamma\alpha_n v(z^\star(X_n)) \right\|^2 = (1 - \alpha_n)^2 \left\| Z_n - z^\star(X_n) \right\|^2 \tag{49a}$$
$$- 2\gamma\alpha_n(1 - \alpha_n)\langle v(z^\star(X_n)), Z_n - z^\star(X_n) \rangle \tag{49b}$$
$$+ \gamma^2\alpha_n^2 \left\| v(z^\star(X_n)) \right\|^2 \tag{49c}$$

Next for the inner product term equation 48b, writing $V_{n+1} = v(Z_n) + \xi_{n+1}$, we have that

$$2\gamma\alpha_n\langle(1-\alpha_n)(Z_n - z^\star(X_n)) - \gamma\alpha_n v(z^\star(X_n)), V_{n+1}\rangle = 2\gamma\alpha_n(1-\alpha_n)\langle v(Z_n), Z_n - z^\star(X_n)\rangle \tag{50a}$$

$$+ 2\gamma\alpha_n\langle\xi_{n+1}, (1-\alpha_n)(Z_n - z^\star(X_n)) - \gamma\alpha_n v(z^\star(X_n))\rangle \tag{50b}$$

$$- 2\gamma_n^2\alpha_n^2\langle v(Z_n), v(z^\star(X_n))\rangle. \tag{50c}$$

We handle each of the terms in equation 50 separately. For equation 50a, we sum with the inner product term equation 49b and apply the monotonicity property of the pseudo-gradient.

$$2\gamma\alpha_n(1-\alpha_n)\langle v(Z_n) - v(z^\star(X_n)), Z_n - z^\star(X_n)\rangle \leq -2\gamma\mu\alpha_n(1-\alpha_n)\|Z_n - z^\star(X_n)\|^2. \tag{51}$$

For the inner product equation 50c, we apply the following identity

$$-2\gamma_n^2\alpha_n^2\langle v(Z_n), v(z^\star(X_n))\rangle = \gamma^2\alpha_n^2\left(\|v(Z_n) - v(z^\star(X_n))\|^2 - \|v(Z_n)\|^2 - \|v(z^\star(X))\|^2\right) \tag{52}$$

We remark that the final term of equation 52 cancels out with the term equation 49c. In addition, the first term of equation 52 satisfies the following inequality owing to the Lipchitz continuity of $v$

$$\gamma^2\alpha_n^2\|v(Z_n) - v(z^\star(X_n))\|^2 \leq \gamma^2 L^2\alpha_n^2\|Z_n - z^\star(X_n)\|^2. \tag{53}$$

Combining equation 49, equation 50, equation 51 equation 52 and equation 53, we have the following inequality for equation 48

$$\|Z_{n+1}^o - z^\star(X_n)\|^2 \leq \left((1-\alpha_n)^2 - 2\gamma\mu\alpha_n(1-\alpha_n) + \gamma^2 L^2\alpha_n^2\right)\|Z_n - z^\star(X_n)\|^2 \tag{54a}$$

$$+ 2\gamma\alpha_n\langle\xi_{n+1}, (1-\alpha_n)(Z_n - z^\star(X_n)) - \gamma\alpha_n v(z^\star(X_n))\rangle \tag{54b}$$

$$- \gamma^2\alpha_n^2\|v(Z_n)\|^2 \tag{54c}$$

$$+ \gamma^2\alpha_n^2\|V_{n+1}\|^2. \tag{54d}$$

Next we take the conditional expectation with respect to $\mathcal{F}_n$ in the inner product tern equation 54b in order to extract the bias.

$$\mathbb{E}_n[2\gamma\alpha_n\langle\xi_{n+1}, (1-\alpha_n)(Z_n - z^\star(X_n)) - \gamma\alpha_n v(z^\star(X_n))\rangle] = 2\gamma\alpha_n(1-\alpha_n)\langle b_{n+1}, Z_n - z^\star(X_n)\rangle \tag{55a}$$

$$- 2\gamma^2\alpha_n^2\langle b_{n+1}, v(z^\star(X_n))\rangle. \tag{55b}$$

For equation 55b, we apply Young's inequality for products, which implies that

$$2\gamma^2\alpha_n^2\langle b_{n+1}, v(z^\star(X_n))\rangle \leq \gamma^2\alpha_n^2\|b_{n+1}\|^2 + \gamma^2\alpha_n^2\|v(z^\star(X_n))\|^2. \tag{56}$$

Similarly, for equation 55a, an application of Young's inequality for products implies

$$2\gamma\alpha_n(1-\alpha_n)\langle b_{n+1}, Z_n - z^\star(X_n)\rangle \leq \gamma\alpha_n(1-\alpha_n)\delta_n\|Z_n - z^\star(X_n)\|^2 + \gamma\alpha_n(1-\alpha_n)\frac{1}{\delta_n}\|b_{n+1}\|^2$$

$$\leq \gamma\alpha_n\delta_n\|Z_n - z^\star(X_n)\|^2 + \frac{\gamma\alpha_n}{\delta_n}\|b_{n+1}\|^2. \tag{57a}$$

Taking the conditional expectation and substituting equation 56 and equation 57 into equation 54, we have that

$$\mathbb{E}_n\|Z_{n+1}^o - z^\star(X_n)\|^2 \leq \left((1-\alpha_n)^2 - 2\gamma\mu\alpha_n(1-\alpha_n) + \gamma^2 L^2\alpha_n^2 + \gamma\alpha_n\delta_n\right)\|Z_n - z^\star(X_n)\|^2 \tag{58a}$$

$$+ (\gamma^2\alpha_n^2 + \frac{\gamma\alpha_n}{\delta_n})\|b_{n+1}\|^2 \tag{58b}$$

$$+ \gamma^2\alpha_n^2\|v(z^\star(X_n))\|^2 - \gamma^2\alpha_n^2\|v(Z_n)\|^2 \tag{58c}$$

$$+ \gamma^2\alpha_n^2\mathbb{E}_n\|V_{n+1}\|^2. \tag{58d}$$

We bound the term equation 58c using the reverse triangle inequality,

$$\gamma^2\alpha_n^2 \left\|v(z^\star(X_n))\right\|^2 - \gamma^2\alpha_n^2 \left\|v(Z_n)\right\|^2 \leq \gamma^2\alpha_n^2 \left\|v(Z_n) - v(z^\star(X_n))\right\|^2 \leq \gamma^2 L^2\alpha_n^2 \left\|Z_n - z^\star(X_n)\right\|^2. \tag{59}$$

This transforms equation 58 into the following

$$\mathbb{E}_n \left\|Z_{n+1}^o - z^\star(X_n)\right\|^2 \leq \left((1-\alpha_n)^2 - 2\gamma\mu\alpha_n(1-\alpha_n) + 2\gamma^2 L^2\alpha_n^2 + \gamma\alpha_n\delta_n\right) \left\|Z_n - z^\star(X_n)\right\|^2 \tag{60a}$$

$$+ (\gamma^2\alpha_n^2 + \frac{\gamma\alpha_n}{\delta_n}) \left\|b_{n+1}\right\|^2 \tag{60b}$$

$$+ \gamma^2\alpha_n^2 \mathbb{E}_n \left\|V_{n+1}\right\|^2. \tag{60c}$$

Let's expand the coefficient of $\left\|Z_n - z^\star(X_n)\right\|^2$ in equation 60

$$(1-\alpha_n)^2 - 2\gamma\mu\alpha_n(1-\alpha_n) + 2\gamma^2 L^2\alpha_n^2 + \gamma\alpha_n\delta_n = 1 - 2(1+\gamma\mu)\alpha_n + (1+2\gamma\mu+2\gamma^2 L^2)\alpha_n^2 + \gamma\alpha_n\delta_n \tag{61}$$

We set $\theta_n = \theta_0\alpha_n$ for some constant $\theta_0 > 0$. Then we expand

$$(1+\theta_n)\left[(1-\alpha_n)^2 - 2\gamma\mu\alpha_n(1-\alpha_n) + 2\gamma^2 L^2\alpha_n^2 + \gamma\alpha_n\delta_n\right] = 1 - [2(1+\gamma\mu) - \theta_0]\alpha_n$$
$$+ (1+2\gamma\mu+2\gamma^2 L^2 - 2\theta_0(1+\gamma\mu))\alpha_n^2$$
$$+ \theta_0(1+2\gamma\mu+2\gamma^2 L^2)\alpha_n^3$$
$$+ \gamma\alpha_n\delta_n(1+\theta_0\alpha_n)$$

By taking $\theta_0 = 1$ we ensure that this coefficient is $\mathcal{O}(1 - (1+2\gamma\mu)\alpha_n)$.

Returning to equation 44, with $\theta_n = \alpha_n$, we arrive at the descent inequality

$$\mathbb{E}_n \left\|Z_{n+1} - z^\star(X_{n+1})\right\|^2 \leq \left(1 - (1+2\gamma\mu-\gamma\delta_n)\alpha_n - (1-2\gamma^2 L^2-\gamma\delta_n)\alpha_n^2 + (1+2\gamma\mu+2\gamma^2 L^2)\alpha_n^3\right) \left\|Z_n - z^\star(X_n)\right\|^2$$
$$+ (1+\alpha_n)(\gamma^2\alpha_n^2 + \frac{\gamma\alpha_n}{\delta_n}) \left\|b_{n+1}\right\|^2$$
$$+ \left(\gamma^2\alpha_n^2(1+\alpha_n) + L_z^2\beta_n^2(1+\frac{1}{\alpha_n})\right)\mathbb{E}_n \left\|V_{n+1}\right\|^2.$$

$\square$

We now use Lemma E.1 in order to obtain an asymptotic descent inequality for the time-rescaled quantity $\frac{\beta_n}{\alpha_n} \left\|Z_n - z^\star(X_n)\right\|^2$.

**Lemma E.2** (Time-Rescaled Fast Descent Inequality). *Under Assumptions 1-2, for all $n$ sufficiently large,*

$$\mathbb{E}_n \frac{\beta_{n+1}}{\alpha_{n+1}} \left\|Z_{n+1} - z^\star(X_{n+1})\right\|^2 \leq \left(1 - (\frac{1}{2} + 2\gamma\mu)\alpha_n.\right)\frac{\beta_n}{\alpha_n} \left\|Z_n - z^\star(X_n)\right\|^2 \tag{64a}$$

$$+ \frac{2\gamma\beta_n}{\delta_n} \left\|b_{n+1}\right\|^2 \tag{64b}$$

$$+ 2\left(\gamma^2\alpha_n\beta_n + L_z^2\frac{\beta_n^3}{\alpha_n^2}\right)\mathbb{E}_n \left\|V_{n+1}\right\|^2. \tag{64c}$$

*Proof.* As a consequence of Assumption 2, for sufficiently large $n$,

$$1 - (1+2\gamma\mu-\gamma\delta_n)\alpha_n - (1-2\gamma^2 L^2-\gamma\delta_n)\alpha_n^2 + (1+2\gamma\mu+2\gamma^2 L^2)\alpha_n^3 \leq 1 - (\frac{1}{2}+2\gamma\mu)\alpha_n. \tag{65}$$

Similarly, for sufficiently large $n$, we have that

$$(1+\alpha_n)(\gamma^2\alpha_n^2 + \frac{\gamma\alpha_n}{\delta_n}) \leq \frac{2\gamma\alpha_n}{\delta_n}. \tag{66}$$

Finally,

$$\gamma^2 \alpha_n^2 (1 + \alpha_n) + L_z^2 \beta_n^2 (1 + \frac{1}{\alpha_n}) \leq 2\gamma^2 \alpha_n^2 + 2L_z^2 \frac{\beta_n^2}{\alpha_n} \tag{67}$$

Applying these inequalities for each coefficient in the descent inequality of Lemma E.1, we have that for sufficiently large n,

$$\mathbb{E}_n \left\| Z_{n+1} - z^\star(X_{n+1}) \right\|^2 \leq \left( 1 - (\frac{1}{2} + 2\gamma\mu)\alpha_n. \right) \left\| Z_n - z^\star(X_n) \right\|^2$$

$$+ \frac{2\gamma\alpha_n}{\delta_n} \left\| b_{n+1} \right\|^2$$

$$+ 2\left( \gamma^2 \alpha_n^2 + L_z^2 \frac{\beta_n^2}{\alpha_n} \right) \mathbb{E}_n \left\| V_{n+1} \right\|^2.$$

Rescaling by the factor $\frac{\beta_{n+1}}{\alpha_{n+1}}$ and applying Assumption 2.2 to the right hand side, we have that, for all $n$ sufficiently large,

$$\mathbb{E}_n \frac{\beta_{n+1}}{\alpha_{n+1}} \left\| Z_{n+1} - z^\star(X_{n+1}) \right\|^2 \leq \left( 1 - (\frac{1}{2} + 2\gamma\mu)\alpha_n. \right) \frac{\beta_n}{\alpha_n} \left\| Z_n - z^\star(X_n) \right\|^2$$

$$+ \frac{2\gamma\beta_n}{\delta_n} \left\| b_{n+1} \right\|^2$$

$$+ 2\left( \gamma^2 \alpha_n \beta_n + L_z^2 \frac{\beta_n^3}{\alpha_n^2} \right) \mathbb{E}_n \left\| V_{n+1} \right\|^2.$$

$\square$

We now utilize the identity of Lemma D.2 in order to obtain a descent inequality for the slow process.

**Lemma E.3** (Slow Descent Inequality). *Suppose Assumptions 1-2 hold and, if the game is strongly monotone, that $\gamma\mu < \frac{1}{4}$. Let $x^\star \in \mathcal{X}^\star$ be a Nash equilibrium. For all $n \geq n_0 := \inf\{m \geq 1 : x^\star \in R_m \mathbb{B}^d\}$,*

$$\mathbb{E}_n \left\| X_{n+1} - x^\star \right\|^2 \leq (1 - \mu\beta_n + \beta_n\delta_n) \left\| X_n - x^\star \right\|^2 \tag{70a}$$

$$+ \frac{1}{\gamma} \left( \frac{2\gamma\mu}{1 - 4\mu\gamma} + \gamma^2 L^2 + 2 \right) \beta_n \left\| Z - z^\star(X_n) \right\|^2 \tag{70b}$$

$$+ \frac{\beta_n}{\delta_n} \left\| b_{n+1} \right\|^2 \tag{70c}$$

$$- \frac{1}{2}\gamma\beta_n \left\| v(Z_n) \right\|^2 - \frac{1}{2}\gamma\beta_n \left\| v(z^\star(X_n)) \right\|^2 \tag{70d}$$

$$+ \beta_n^2 \mathbb{E}_n \left\| V_{n+1} \right\|^2. \tag{70e}$$

*Proof.* Suppose that $x^\star \in \mathcal{X}^\star$ is a Nash equilibrium and suppose that $n$ is sufficiently large so that $x^\star \in R_n \mathbb{B}^d$. As a consequence of the non-expansiveness of the projection operator,

$$\left\| X_{n+1} - x^\star \right\|^2 = \left\| \text{Proj}_{R_{n+1}\mathbb{B}^d} X_{n+1}^o - \text{Proj}_{R_{n+1}\mathbb{B}^d} x^\star \right\|^2 \tag{71}$$

$$\leq \left\| X_{n+1}^o - x^\star \right\|^2 \tag{72}$$

$$= \left\| X_n - x^\star + \beta_n V_{n+1} \right\|^2 \tag{73}$$

$$= \left\| X_n - x^\star \right\|^2 + 2\beta_n \langle V_{n+1}, X_n - x^\star \rangle + \beta_n^2 \left\| V_{n+1} \right\|^2. \tag{74}$$

Let us express the gradient estimate $V_{n+1} = v(Z_n) + \xi_{n+1}$. We may then take the conditional expectation and expand the inner product term of equation 74 as follows:

$$\mathbb{E}_n 2\beta_n \langle V_{n+1}, X_n - x^\star \rangle = 2\beta_n \langle v(Z_n), X_n - x^\star \rangle + 2\beta_n \langle \mathbb{E}_n \xi_{n+1}, X_n - x^\star \rangle. \tag{75}$$

For the bias term, we apply Young's inequality for products,

$$2\beta_n \langle \mathbb{E}_n \xi_{n+1}, X_n - x^\star \rangle \leq \beta_n \delta_n \left\| X_n - x^\star \right\|^2 + \frac{\beta_n}{\delta_n} \left\| b_{n+1} \right\|^2. \tag{76}$$

For the remaining term, we may apply the result of Lemma D.2 in order to obtain

$$2\beta_n \langle v(Z_n), X_n - x^\star \rangle \leq - \mu\beta_n \|X_n - x^\star\|^2 \tag{77a}$$

$$+ \frac{1}{\gamma}\left(\frac{2\gamma\mu}{1-4\mu\gamma} + \gamma^2 L^2 + 2\right)\beta_n \|Z - z^\star(X_n)\|^2 \tag{77b}$$

$$- \frac{1}{2}\gamma\beta_n \|v(Z_n)\|^2 - \frac{1}{2}\gamma\beta_n \|v(z^\star(X_n))\|^2 \tag{77c}$$

Taking the conditional expectation in equation 74 and applying the inequalities equation 76 and equation 77, we arrive at the claimed descent inequality, equation 70 $\qquad\square$

# F   CONVERGENCE PROOFS

## F.1   PROOF OF PROPOSITION 4.4

*Proof of Proposition 4.4.* We have that $\|b_{n+1}\| = \mathcal{O}(\delta_n)$ as a property of the SPSA gradient estimate. Moreover, Lemma 3.1 states that $\|V_{n+1}\| = \mathcal{O}(R_n)$. With these estimates for the error terms, we rewrite the descent inequality of E.2 in the following form:

$$\mathbb{E}_n \frac{\beta_{n+1}}{\alpha_{n+1}} \|Z_{n+1} - z^\star(X_{n+1})\|^2 \leq \left(1 - (\frac{1}{2}+2\gamma\mu)\alpha_n\right)\frac{\beta_n}{\alpha_n} \|Z_n - z^\star(X_n)\|^2 \tag{78a}$$

$$+ \mathcal{O}\left(\beta_n\delta_n + \alpha_n\beta_n R_n^2 + \frac{\beta_n^3 R_n^2}{\alpha_n^2}\right) \tag{78b}$$

As a result of Assumption 2, we have that the conditions of Theorem B.2 are satisfied, hence

$$\frac{\beta_n}{\alpha_n} \|Z_n - z^\star(X_n)\|^2 \to 0 \text{ a.s. and in expectation.} \tag{79}$$

Suppose, in addition, Assumption 3 holds. Taking the expectation in equation 78, we obtain the following:

$$\mathbb{E}\frac{\beta_{n+1}}{\alpha_{n+1}} \|Z_{n+1} - z^\star(X_{n+1})\|^2 \leq \left(1 - (\frac{1}{2}+2\gamma\mu)\frac{\alpha}{n^a}\right)\mathbb{E}\frac{\beta_n}{\alpha_n} \|Z_n - z^\star(X_n)\|^2 \tag{80a}$$

$$+ \mathcal{O}\left(\frac{(\log n)^2}{n^{b+d}} + \frac{(\log n)^2}{n^{a+b}} + \frac{(\log n)^2}{n^{3b-2a}}\right) \tag{80b}$$

Since $\log n = \mathcal{O}(n^{\frac{1}{2}\epsilon})$ for any $\epsilon > 0$, we have that equation 80 implies that for any $\epsilon > 0$, for all $n$ sufficiently large,

$$\mathbb{E}\frac{\beta_{n+1}}{\alpha_{n+1}} \|Z_{n+1} - z^\star(X_{n+1})\|^2 \leq \left(1 - (\frac{1}{2}+2\gamma\mu)\frac{\alpha}{n^a}\right)\mathbb{E}\frac{\beta_n}{\alpha_n} \|Z_n - z^\star(X_n)\|^2 \tag{81a}$$

$$+ \mathcal{O}\left(\frac{1}{n^{e+a-\epsilon}}\right) \tag{81b}$$

where $e + a = \min\{b+d, a+b, 3b-2a\}$.

As a consequence of the summability constraints of Assumption 2, we have that $\min\{b + d - a, b, 3b - 3a\} > 0$. Hence, if we set $0 < \epsilon < \min\{b + d - a, b, 3b - 3a\}$, then we have that $e - \epsilon > 0$. An application of Chung's lemma B.3 (noting that $a < 1$) implies that

$$\mathbb{E}\frac{\beta_n}{\alpha_n} \|Z_n - z^\star(X_n)\|^2 = \mathcal{O}\left(\frac{1}{n^{e-\epsilon}}\right). \tag{82}$$

$\qquad\square$

## F.2   PROOF OF PROPOSITION 4.5

*Proof of Proposition 4.5.* We begin by showing that the positive terms in the descent inequality equation 70 of Lemma E.3 are finite in series. First, we apply Proposition 4.4 in order to obtain the following estimate for any $0 < \epsilon < \min\{b + d - a, b, 3b - 3a\}$ and sufficiently large $n$

$$\beta_n \mathbb{E}\left[\|Z_n - z^\star(X_n)\|^2\right] = \mathcal{O}\left(\frac{1}{n^{e+a-\epsilon}}\right), \tag{83}$$

where $e + a - \epsilon = \min\{b + d, a + b, 3b - 2a\} - \epsilon$.

As a consequence of the summability conditions of Assumption 2, we have that $\min\{b + d, a + b, 3b - 2a\} > 1$ and so we may set $0 < \epsilon < \min\{b + d - 1, a + b - 1, 3b - 2a - 1\}$. In which case, we have that $e + a - \epsilon > 1$. Hence we have that

$$\sum_{n=1}^{\infty} \beta_n \mathbb{E}\left[\|Z_n - z^\star(X_n)\|^2\right] < +\infty. \tag{84}$$

It is the case that $\|b_{n+1}\| = \mathcal{O}(\delta_n)$. Hence $\frac{\beta_n}{\delta_n}\|b_{n+1}\|^2 = \mathcal{O}(\delta_n\beta_n)$, which is finite in series owing to Assumption 2.

Similarly, as a consequence of Lemma 3.1, we have that $\beta_n^2 \mathbb{E}_n \|V_{n+1}\|^2 = \mathcal{O}(\beta_n^2 R_n^2)$. Since $\beta_n = o(\alpha_n)$ and by Assumption 2, $\sum_{n=1}^{\infty} \alpha_n\beta_n R_n^2 < \infty$, we have that $\sum_{n=1}^{\infty} \beta_n^2 R_n^2 < \infty$ and $\beta_n^2 \mathbb{E}_n \|V_{n+1}\|^2$ is finite in series.

With each of the positive terms in the descent inequality equation 70 of Lemma E.3 are finite in series, applying the stochastic approximation theorem of Robbins-Seigmund B.1, we have that $\|X_n - x^\star\|^2$ converges almost surely to finite random variable, and that the negative terms satisfy

$$\sum_{n=1}^{\infty} (\frac{1}{2}\gamma\beta_n \|v(Z_n)\|^2 + \frac{1}{2}\gamma\beta_n \|v(z^\star(X_n))\|^2) < \infty \text{ a.s.} \tag{85}$$

In particular, since $\sum_{n=1}^{\infty} \beta_n = +\infty$, it must be the case that

$$\liminf_{n\to\infty}(\|v(Z_n)\|^2 + \|v(z^\star(X_n))\|^2) = 0. \tag{86}$$

$\square$

## F.3 Proof of Theorem 4.1

*Proof of Theorem 4.1.* Let $n_k$ be a sequence satisfying $\|v(Z_{n_k})\|^2 + \|z^\star(X_{n_k})\|^2 \to 0$. In particular we note that

$$v(Z_{n_k}) \to 0 \text{ and } v(z^\star(X_{n_k})) \to 0.$$

Let $x^\star \in \mathcal{X}^\star$. The almost sure convergence of $\|X_{n_k} - x^\star\|$ implies that $X_{n_k}$ is almost surely bounded. In particular there exists a subsequence $X_{n_{k_j}}$ converging to a limit $x_\infty \in \mathbb{R}^D$. For the sake of notation, we will adopt the convention that the subsubsequence $n_{k_j}$ corresponds to an increasing function $\omega : \mathbb{N} \to \mathbb{N}$. By Proposition 4.4,

$$\left\|Z_{\omega(n)} - z^\star(X_{\omega(n)})\right\| \to 0 \text{ a.s.}$$

Expanding $z^\star$, by the fixed point equation, we have that $X_{\omega(n)} = z^\star(X_{\omega(n)}) - \gamma v(z^\star(X_{\omega(n)}))$. An application of the triangle inequality yields that

$$\left\|Z_{\omega(n)} - X_{\omega(n)}\right\| \le \left\|Z_{\omega(n)} - z^\star(X_{\omega(n)})\right\| + \gamma \left\|v(z^\star(X_{\omega(n)}))\right\| \to 0 \text{ a.s.}$$

This implies that $Z_{\omega(n)} \to x_\infty$ a.s. and by the continuity of $v$,

$$v(x_\infty) = \lim_{n\to\infty} v(Z_{\omega(n)}) = 0.$$

This is precisely that $x_\infty \in \mathcal{X}^\star$. Taking $x^\star = x_\infty$, the almost sure convergence of $\|X_n - x_\infty\|$ to a finite random variable and the convergence of the subsequence $\left\|X_{\omega(n)} - x_\infty\right\|$ to zero, implies that the entire sequence $X_n \to x_\infty \in \mathcal{X}^\star$ a.s. $\square$

## F.4 Proof of Theorem 4.2

*Proof of Theorem 4.2.* For all $n$ sufficiently large, as a consequence of Assumption 2,

$$1 - \mu\beta_n + \delta_n\beta_n \le 1 - \frac{1}{2}\mu\beta_n. \tag{87}$$

Taking the expectation in the descent inequality 70, we have that for all $n$ sufficiently large,

$$\mathbb{E}\left\|X_{n+1} - x^\star\right\|^2 \leq \left(1 - \frac{1}{2}\mu\beta_n\right)\mathbb{E}\left\|X_n - x^\star\right\|^2 \tag{88a}$$

$$+ \frac{1}{\gamma}\left(\frac{2\gamma\mu}{1 - 4\mu\gamma} + \gamma^2 L^2 + 2\right)\beta_n\mathbb{E}\left\|Z - z^\star(X_n)\right\|^2 \tag{88b}$$

$$+ \mathcal{O}\left(\beta_n\delta_n + \beta_n^2 R_n^2\right). \tag{88c}$$

We apply Proposition 4.4 in order to obtain the following estimate for any $0 < \epsilon < \min\{b + d - 1, a + b - 1, 3b - 2a - 1\}$ and for sufficiently large $n$,

$$\beta_n\mathbb{E}\left[\left\|Z_n - z^\star(X_n)\right\|^2\right] = \mathcal{O}\left(\frac{1}{n^{e+a-\epsilon}}\right), \tag{89}$$

where $e + a - \epsilon = \min\{b + d, a + b, 3b - 2a\} - \epsilon$. Since the parameter sequences take the form of Assumption 3, we may rewrite 88 as follows:

$$\mathbb{E}\left\|X_{n+1} - x^\star\right\|^2 \leq \left(1 - \frac{\beta\mu}{2n^b}\right)\mathbb{E}\left\|X_n - x^\star\right\|^2 \tag{90a}$$

$$+ \mathcal{O}\left(\frac{(\log n)^2}{n^{b+d}} + \frac{(\log n)^2}{n^{2b}} + \frac{1}{n^{e+a-\epsilon}}\right). \tag{90b}$$

Again, noting that $\log n = \mathcal{O}(n^{\frac{1}{2}\epsilon})$, we have that, for all $n$ sufficiently large,

$$\mathbb{E}\left\|X_{n+1} - x^\star\right\|^2 \leq \left(1 - \frac{\beta\mu}{2n^b}\right)\mathbb{E}\left\|X_n - x^\star\right\|^2 \tag{91a}$$

$$+ \mathcal{O}\left(\frac{1}{n^{b+d-\epsilon}} + \frac{1}{n^{2b-\epsilon}} + \frac{1}{n^{e+a-\epsilon}}\right). \tag{91b}$$

Let $f := \min\{d, b, a - b\}$. Noting that $e + a - b = \min\{d, a, 2b - 2a\}$, we have that

$$f = \min\{d, a, 2b - 2a\}. \tag{92}$$

We may choose $0 < \epsilon < \min\{a, 2b - 2a, 3b - 2a - 1, a + b - 1\}$ so that both the following hold: $e + a - \epsilon > 1$ and $f - \epsilon > 0$. This enables us to rewrite equation 91 in the following form

$$\mathbb{E}\left\|X_{n+1} - x^\star\right\|^2 \leq \left(1 - \frac{\beta\mu}{2n^b}\right)\mathbb{E}\left\|X_n - x^\star\right\|^2 \tag{93}$$

$$+ \mathcal{O}\left(\frac{1}{n^{f+b-\epsilon}}\right), \tag{94}$$

and an Application of Chung's Lemma B.3 yields that

$$\mathbb{E}\left\|X_n - x^\star\right\|^2 \leq \mathcal{O}\left(\frac{1}{n^{f-\epsilon}}\right), \tag{95}$$

Since $\epsilon > 0$ can be taken to be arbitrarily small, we have that

$$\mathbb{E}\left\|X_n - x^\star\right\|^2 \leq \tilde{\mathcal{O}}\left(\frac{1}{n^f}\right), \tag{96}$$

as claimed. $\qquad\square$

## G EXPERIMENTAL SETTING FOR SECTION 5

The experiments in Section 5 were conducted in macOS 14.5 with Apple M2 Max and 32GB of RAM.

In Figure 2c is a log-log graph comparing the average norm of the iterates, averaged over 10 instances with random seeds. The shaded region corresponds to the 25-75 percentile of the ensemble. In each instance, we initialize the game with action $x_0 = (10, 20)$. The parameter sequences are tuned as follows:

- SPOG: $\gamma = \frac{1}{2}$, $\alpha_n = (\frac{50}{50+n})^{0.66}$, $\beta_n = \frac{100}{100+n}$, $\delta_n = (\frac{1000}{1000+n})^{0.66}$ and $R_n = 100\log(n+1)$,

- OG+: $\gamma_n = 0.1(\frac{1000}{n+1000})^{0.25}$, $\eta_n = 0.1(\frac{1}{n+1})^{0.5}$,

- OG+SPSA: $\gamma = (\frac{50}{50+n})^{0.66}$, $\eta = \frac{100}{100+n}$ and $\delta_n = (\frac{1000}{1000+n})^{0.66}$.

- (OG): $\gamma_n = 0.1(\frac{1}{n+1})^{0.5}$.

In keeping with Theorem 4.2 and Remark 4.1, we choose $a = d = 0.66 \approx \frac{2}{3}, b = 1$ in order to approximate the best-rate attained through our analysis. In addition, in order to prevent the parameter sequences from decaying too quickly, we opt to translate $n$ and rescale the parameter sequences, as above; this ensures the trajectories of the iterates are sufficiently long. The parameters for OG+ with additive noise reflect the constraints stated in the last-iterate convergence result in Hsieh et al. (2022). The parameters for OG are taken from a similar experiment in Hsieh et al. (2022).

In terms of noise models for OG and OG+, we consider a normal $\mathcal{N}(0, 0.2)$ distribution on the noise $\xi_{n+1}$ which is *additive*, that is, $\hat{v}_{n+1,i} = v_i(x_{n+\frac{1}{2}}) + \xi_{n+1,i}$ for $i = 1, 2$.

# H SUPPORTING LEMMAS

We state the following Lemma concerning inner product spaces, which we frequently make use of and refer to as Young's inequality.

**Lemma H.1.** *Let $(\mathbb{R}^D, \langle \cdot, \cdot \rangle)$ be an inner product space with induced norm $\|\cdot\|$. For any $\theta > 0$, and any $x, y \in \mathbb{R}^D$, the following hold:*

*(i)*
$$\langle x, y \rangle \leq \frac{\theta}{2}\|x\|^2 + \frac{1}{2\theta}\|y\|^2,$$

*(ii)*
$$\|x + y\|^2 \leq (1+\theta)\|x\|^2 + (1 + \frac{1}{\theta})\|y\|^2.$$

*Proof.* We exploit the bi-linearity of the inner product to write

$$\langle x, y \rangle = \langle \sqrt{\theta}x, \frac{y}{\sqrt{\theta}} \rangle. \tag{97}$$

Expanding the norm, we next observe that

$$0 \leq \left\|\sqrt{\theta}x - \frac{y}{\sqrt{\theta}}\right\|^2 \leq \theta\|x\|^2 + \frac{1}{\theta}\|y\|^2 - 2\langle \sqrt{\theta}x, \frac{y}{\sqrt{\theta}} \rangle \tag{98}$$

Rearranging equation 98 and applying equation 97, we arrive at the claimed result (i).

We see that (i) $\implies$ (ii) when the expand the following inner product

$$\|x + y\|^2 = \|x\|^2 + \|y\|^2 + 2\langle x, y \rangle.$$

$\square$

