# OpenReview forum: "Simultaneously Perturbed Optimistic Gradient Methods for Payoff-Based Learning in Games"
_ICLR.cc/2026/Conference — Submitted to ICLR 2026_

### Official Review · Reviewer_DYPv · 2025-10-14

**Soundness:** 2
**Presentation:** 2
**Contribution:** 2
**Rating:** 2
**Confidence:** 3

**Summary:**

This paper introduces a new algorithm, SPOG, designed to solve strongly monotone games under bandit (zeroth-order) feedback. The algorithm achieves a convergence rate of $\tilde{O}(n^{-2/3})$, improving upon the rate attained in previous work.

**Strengths:**

- SPOG obtains a faster convergence rate compared to the previous method (Table 1).
- The empirical evidence shows that SPOG is faster than the baselines.

**Weaknesses:**

The big-O notation in the main theorem (Theorem 4.2) conceals factors that are **exponentially large**. Lemma 3.1 holds only for sufficiently large $n$, and an examination of its proof shows that this threshold $n$ is itself exponentially large. Specifically, in line 709, it requires $\Theta(\frac{n^{a-d}}{\log n})=\frac{\alpha_{n-1} R_n}{\delta_n} \leq \min( \frac{1}{4M\sqrt{N}}, \frac{1}{2\gamma M} )$, while the optimal convergence rate is obtained when $a=d$ (Theorem 4.2).
Consequently, this exponential dependence propagates into the convergence rate stated in Theorem 4.2, which relies on Lemma 3.1.

Furthermore, I recommend that the authors **explicitly specify** the required $n$ in Lemma 3.1 rather than hiding it within the asymptotic notation. Since the main claim of the paper concerns achieving a faster rate, even the constants involved are significant and should be made transparent.

**Questions:**

Given the exponentially large factor implicated in Theorem 4.2, I encourage the authors to evaluate SPOG on a broader suite of (larger) games and to include additional baselines. Without such expanded empirical evidence, the current experiments are insufficient to substantiate the claimed efficiency of SPOG.

---

> ### Author Response · Authors · 2025-11-21
> **Reply 1/2**
>
> Dear reviewer,
>
> Thank you for your time and input. We reply to your remarks and questions below:
>
> ---
> > The big-O notation in the main theorem (Theorem 4.2) conceals factors that are exponentially large.
>
> The rates we provide are indeed asymptotic (we never claimed otherwise), and they are compared to *other asymptotic rates* in the literature.
>
> Specifically, our analysis has been carefully calibrated to optimize the dependence of the derived rate on $n$, without any further considerations. This is in line with the relevant literature on the topic; using Table 1 as a reference point, we have:
> - Dong et al (2025): bandit feedback, $\mathcal{O}(n^{-1/4})$ rate, ***asymptotic***.
> - Bravo et al (2018): bandit feedback, $\mathcal{O}(n^{-1/3})$ rate, ***asymptotic***.
> - Tatarenko \& Kamgarpour (2024): bandit feedback, $\mathcal{O}(n^{-1/2})$, ***asymptotic***.
>
> The *only* non-asymptotic rate in the case of learning with bandit feedback is the $\mathcal{O}(n^{-1/2})$ rate of Drusvyatskiy et al. (2022). However, this result applies only to learning over *bounded* domains—and the lack of boundedness is one of the main reasons that our analysis is asymptotic instead of anytime.
> <!-- The other works surveyed in Table 1 (Cai and Zheng, 2023; Abe et al., 2025) both rely on the availability of first-order gradient-based feedback (not payoff-based), and this plays a crucial role in obtaining anytime bounds (since the variance of the estimator is a priori bounded—or zero in the case of learning with perfect gradient feedback). -->
>
> So, if we are to make a bona fide, "apples-to-apples" comparison to the relevant bandit learning results in the literature, please note that, except for the work of Drusvyatskiy et al, all the rates provided in the surveyed works are also asymptotic.
>
>
> > Lemma 3.1 holds only for sufficiently large , and an examination of its proof shows that this threshold  is itself exponentially large. Specifically, in line 709, it requires $\Theta(n^{a-d}/\log n) \leq [\textrm{a constant}]$, while the optimal convergence rate is obtained when $a=d$ (Theorem 4.2). Consequently, this exponential dependence propagates into the convergence rate stated in Theorem 4.2, which relies on Lemma 3.1.
>
> The variance bound of Lemma 3.1 activates for all $n\geq n_1$, where
> $$
> n_1 = \sup_{k\geq 1} \left[
> \frac{\alpha_{k-1} R_k}{\delta_k} > \frac{\min\{1, 2\sqrt{N}/\gamma\}}{16DGN}\right].
> $$
> If SPOG is run with $a=d=2/3$ and $b=1$ as per Assumption 3, we have $\alpha_{n-1} R_n/\delta_n = \mathcal{O}(1/\log n)$, so the number of rounds until Lemma 3.1 binds may indeed be exponential in $D, G$ and $N$. The tuning $a=d=2/3$ has been chosen to optimize the dependence of the derived rate in $n$, without further considerations—as per the likewise asymptotic analysis of Bravo et al (2018), Tatarenko \& Kamgarpour (2024), and Dong et al (2025), upon which it improves.
>
> If the fact that the worst-case dependence on $n_1$ may be exponential in $D,G$ and $N$ is cause for concern, note that it can be avoided by taking the more conservative tuning schedule
> $$
> \alpha_n = \frac{\alpha}{n^{2/3}}
> \qquad
> \beta_n = \frac{\beta}{n}
> \qquad
> \delta_n = \frac{\delta}{n^{2/3-\epsilon}}
> \quad
> \text{ and }
> \quad
> R_n = R \log n.
> $$
> where $\epsilon > 0$ is an arbitrary small constant (e.g., $\epsilon = 1/60$). In this case, **the variance bound of Lemma 3.1 binds after $n_1= \mathrm{poly}(D, G, N, \gamma)$ iterations,** at the cost of only a slight deterioration in the algorithm's convergence rate, namely to
> $$
> \mathbb{E}[\|X_n - x^\ast\|^2]
>     = \mathcal{\tilde O}(n^{-2/3+\epsilon})
> $$
> This rate does not carry an exponential dependence on $N,D$ or $G$, and it still improves on all the surveyed rates—e.g., for $\epsilon = 1/20$, we get an $\mathcal{\tilde O}(n^{-13/20})$ convergence rate. The precise calibration of the "sweet spot" in these trade-offs is obviously application-specific, so it lies beyond the scope of our work.
>
> We only seek to stress again that, to the best of our knowledge, ***our paper provides the first rates for bandit learning in unbounded domains, asymptotic or anytime***. We do not claim that they cannot be finetuned further, but the purpose of our paper is to bring to light a novel series of techniques for a problem that had remained uncracked in the literature until now—and, in this regard, we did not want to encumber the analysis with an additional layer of fine-tuning.
>
> We included all of the above details in the uploaded revision; we hope and trust that this alleviates your concerns on the issue.
>
> ---
>
> > Furthermore, I recommend that the authors explicitly specify the required $n$ in Lemma 3.1 rather than hiding it within the asymptotic notation. Since the main claim of the paper concerns achieving a faster rate, even the constants involved are significant and should be made transparent.
>
> Done—we have included the relevant numerical bounds and details in the uploaded revision (highlighted in blue for your convenience).

---

> > ### Author Response · Authors · 2025-11-21
> > **Reply 2/2**
> >
> > > Given the exponentially large factor implicated in Theorem 4.2, I encourage the authors to evaluate SPOG on a broader suite of (larger) games and to include additional baselines. Without such expanded empirical evidence, the current experiments are insufficient to substantiate the claimed efficiency of SPOG.
> >
> > Please note that the claimed efficiency is in relation to other asymptotic rates in the literature, cf. our answer above and the bandit learning entries of Table 1 in our paper. Beyond this, the purpose of our paper is to bring to light a series of novel **theoretical** techniques for a problem that had remained uncracked in the literature until now (bandit learning in games with unbounded domains). The surveyed rates in the literature all concern games with ***bounded domains*** and the proposed algorithms are not stable when deployed in an unbounded domain. We believe that this would reflect unfairly on the surveyed baselines (which are not designed for problems with unbounded domains), so we did not include such a comparison in the uploaded revision.
> >
> > ---
> >
> > Thank you again for your time.
> >
> > Kind regards,
> >
> > The authors

---

> > > ### Comment · Reviewer_DYPv · 2025-11-24
> > >
> > > Could you explain why Bravo et al. (2018) is classified as asymptotic convergence? Theorem 5.2 is indeed stated using big-O notation, but it seems to me that the proof in Bravo et al. (2018) actually implies an any-time convergence.

---

> > > > ### Author Response · Authors · 2025-11-24
> > > >
> > > > Using the [arxiv version](https://arxiv.org/pdf/1810.01925) of Bravo et al. (2018) as a reference point, the rate in question relies on Chung's lemma (Lemma D.2), which gives a convergence rate of $(Q/R)/n^q$ up to a subleading $o(1/n^q)$ term: the proof of Theorem 5.2 invokes said lemma to get a rate of the form $D_n \leq A/n^{1/3} + o(1/n^{1/3})$.
> > > >
> > > > Bravo et al. do not provide an estimate of the magnitude of the $o(1/n^{1/3})$ term, so their rate is asymptotic.

---

> > > > > ### Comment · Reviewer_DYPv · 2025-11-26
> > > > >
> > > > > Thank you for your response. I will raise my score to 4, and I will further adjust my rating based on the discussion with the other reviewers.

---

### Official Review · Reviewer_YVQY · 2025-10-29

**Soundness:** 3
**Presentation:** 3
**Contribution:** 3
**Rating:** 6
**Confidence:** 3

**Summary:**

This paper studies payoff-based learning in monotone games. In each round, each player observes only zeroth-order (bandit) feedback of their own payoff and cannot observe others. The action space is assumed to be unconstrained (unbounded). The authors propose a novel algorithm, SPOG. It has two main components: (i) an adjusted SPSA gradient estimator that controls variance by using the previous payoff as a baseline, and (ii) a simultaneously perturbed optimistic-gradient update that adapts the OG+ style separation of learning rates to the payoff-based setting where gradients must be estimated. For monotone games, the iterates converge asymptotically to a Nash equilibrium; for strongly monotone games, the authors prove a last-iterate convergence rate of $n^{-2/3}$.

**Strengths:**

- A new last-iterate convergence-rate result for payoff-based learning in monotone games.
- The combination of learning-rate separation with an adjusted gradient estimator is technically interesting and the associated analysis is nontrivial (at least for me).

**Weaknesses:**

- Regarding rates in payoff-based learning, additional related work and comparisons to known lower bounds are needed. For example:
  Fiegel, Côme, et al., "The Harder Path: Last Iterate Convergence for Uncoupled Learning in Zero-Sum Games with Bandit Feedback," ICML.
  Cai, Yang, et al., "Uncoupled and Convergent Learning in Two-Player Zero-Sum Markov Games with Bandit Feedback," NeurIPS 2023.
  Fiegel et al. appear to show an $n^{-1/4}$ lower bound.

- Can the results be extended to the constrained setting? If not directly, what are the main obstacles?

- The input $\gamma$ appears to require knowledge of game-dependent constants $L, G$ (and possibly $N$). Discussion of robustness or adaptive tuning would help.

- Can you obtain a rate for the merely monotone case ($\mu=0$)? One possibility is to add a player-wise regularizer to make the game strongly monotone and then anneal the regularization strength with $n$; can this yield a rate?

- The experimental section focuses on very simple cases. For the two-player zero-sum bilinear game, comparisons with other payoff-based methods (e.g., Fiegel et al.) would strengthen the empirical evidence.

**Questions:**

1. Do you believe the $n^{-2/3}$ rate is tight? Is it possible to provide a lower bound?
2. How does your setting differ from the bandit convex optimization (BCO) model of Shamir (2013), where $\Omega(n^{-1/2})$ lower bounds (for any algorithm) are known? Clarifying the modeling/algorithmic differences and the lower-bound constructions would help reconcile how one might "break" the $n^{-1/2}$ barrier. Does your approach imply fast rates in BCO as well, or do crucial assumptions differ?
3. Can your method be extended to two-point zeroth-order or stochastic first-order settings?
4. How does the rate depend on the strong monotonicity parameter $\mu>0$? From the proof of Proposition 4.4, it superficially looks like the argument might go through with $\mu=0$; where exactly do you use $\mu>0$?
5. In the experiments, can you compare against other payoff-based methods for two-player zero-sum bilinear games?

Minor: Line 105: What is $\mathcal{V}$?

---

> ### Author Response · Authors · 2025-11-21
> **Reply 1/3**
>
> Dear reviewer,
>
> Thank you for your constructive input and positive evaluation! We reply to your remarks and questions below:
>
> ---
>
> > Regarding rates in payoff-based learning, additional related work and comparisons to known lower bounds are needed. For example:
> Fiegel, Côme, et al., "The Harder Path: Last Iterate Convergence for Uncoupled Learning in Zero-Sum Games with Bandit Feedback," ICML.
> Cai, Yang, et al., "Uncoupled and Convergent Learning in Two-Player Zero-Sum Markov Games with Bandit Feedback," NeurIPS 2023.
> Fiegel et al. appear to show an $n^{-1/4}$ lower bound.
>
> Thank you for bringing these additional works to our attention. We have already included them in the review of related work in the uploaded revision.
>
> In more detail, it is true that Fiegel et al. obtain a $\Omega(n^{-1/4})$ lower bound for any uncoupled, zeroth-order learning algorithm across all two-player zero-sum matrix games. However, this does not contradict our upper bound of $\mathcal{O}(n^{-2/3})$ for a number of reasons:
> - The matrix games considered by Fiegel et al. are all constrained.
> - The are not strongly monotone.
> - In finite games, players cannot sample their mixed payoffs at any point of their domain—only at vertices (pure strategies). This means that SPSA-type estimators cannot be employed in that setting (and the variance characteristics of importance-weighted bandit estimators versus SPSA estimators are drastically different).
>
> For all these reasons, a lot of care is required when comparing results for continuous games and finite games. Especially for zero-sum bimatrix games, the bilinearity of the players' payoff function induces a condition known as "metric subregularity" (or "error-bound" in the language of Cheng), which could be leveraged further to obtain sharper rates of convergence, even though 2PZS games are not strongly monotone.
>
> ---
>
> > Can the results be extended to the constrained setting? If not directly, what are the main obstacles?
>
> Yes, we believe that SPOG can be adapted to a constrained setting, resulting in a two-timescales regularized learning algorithm if the domain of the game is bounded. This would remove some challenges (the need for the "projection envelope"), but also introduce others: non-vanishing gradients at boundary solutions, a constrained ODE, the non-linearity of the projection steps that would be required near a solution, etc. All in all, even though we do not see any insurmountable obstacles in adapting SPOG to a constrained setting, the analysis itself would most likely end up being significantly different, and would almost certainly require a paper in itself.
>
> ---
>
> > The input $\gamma$ appears to require knowledge of game-dependent constants $L, G$ (and possibly $N$). Discussion of robustness or adaptive tuning would help.
>
> The upper bound on $\gamma$ is indeed game-dependent, and it reflects the traditional tuning of gradient-based extrapolation methods in the literature (extra-gradient, optimistic gradient, etc.), where the algorithm's (constant) step-size is upper bounded by $2/L$ (or $1/L$ depending on the exact algorithm under study). There is an extensive literature on adaptive / parameter-free methods to overcome the reliance on bounds of this sort via AdaGrad-like step-sizes (a good starting point is the monograph by Orabona); however, there are no results of this type in the bandit, payoff-based setting.
>
> Instead of such an approach, a natural way to circumvent this requirement would be to consider a variable $\gamma_t \propto 1/t^c$ parameter tuning, which would ensure that $\gamma_t$ converges to $0$ on a third timescale, tuned to be slower than all the others. While it is possible to get rid of the upper bound for $\gamma$ in this way, this would introduce an extra layer of complication in the algorithm and its analysis, which we opted to avoid in order to streamline our presentation (the analysis already has several moving parts, so the inclusion of yet another could increase clutter to an undesirable extent).
>
> In the uploaded revision, we included a remark along the above lines (though not the entire discussion, due to space constraints).
>
> ---

---

> > ### Author Response · Authors · 2025-11-21
> > **Reply 2/3**
> >
> > > Can you obtain a rate for the merely monotone case ($\mu=0$)? One possibility is to add a player-wise regularizer to make the game strongly monotone and then anneal the regularization strength with $n$; can this yield a rate?
> >
> > This is a very natural idea, but the details are not simple–essentially because, unless the game has a lot of structure (e.g., like that of a finite game), the distance between the "regularized" equilibrium and the actual equilibrium is not easy to control. We are actively pursuing this idea as future work on the topic.
> >
> > ---
> >
> > > The experimental section focuses on very simple cases. For the two-player zero-sum bilinear game, comparisons with other payoff-based methods (e.g., Fiegel et al.) would strengthen the empirical evidence.
> >
> > Our contributions are primarily theoretical and we include a simple example of a game purely as an illustration of the algorithm's convergence. With that said, we have included an example of a strongly monotone game in the final version. Another challenge is that the majority of the literature on payoff-based methods for learning in monotone games assumes constrained action spaces and regularizes accordingly (as is the case for Fiegel et al) so it is difficult to make a bona fide, "apples-to-apples" comparison with SPOG.
> >
> > ---
> >
> > > Do you believe the $n^{-2/3}$ rate is tight? Is it possible to provide a lower bound?
> >
> > Good question—it is not clear to us. In our case, the main slow-down is due the summability condition $\sum_{n=1}^\infty \beta_n^3\alpha_n^{-2} < \infty$ for SPOG. It is possible that a more refined moment-based analysis could yield a sharper rate, but most of the inequalities involved are fairly tight, so it is unclear what such an analysis would look like. We plan on investigating this issue further in future work—thanks again for the question.
> >
> > ---
> >
> > > How does your setting differ from the bandit convex optimization (BCO) model of Shamir (2013), where $\Omega(n^{-1/2})$ lower bounds (for any algorithm) are known? Clarifying the modeling/algorithmic differences and the lower-bound constructions would help reconcile how one might "break" the $n^{-1/2}$  barrier. Does your approach imply fast rates in BCO as well, or do crucial assumptions differ?
> >
> > The lower bound of Shamir (2013) assumes a gradient estimator of the form form $\hat{V_n}=\frac{d}{\delta_n}u(x_n+\delta_n w_{n+1}) w_{n+1}$. Our chosen estimator $\hat{V_n}=\frac{d}{\delta_n}(u(x_n+\delta_n w_{n+1}) -u(x_{n-1}+\delta_{n-1} w_n) )w_{n+1}$ does not satisfy this assumption and so is not subject to the same lower bound—even though, of course, it remains a “one-point zeroth-order” estimator since, at each iteration, every player makes *only one payoff observation*. We have included a detailed discussion of this point in the uploaded revision.
> >
> > Beyond this, we believe that the reuse of past payoff information as a means to control the variance of SPSA-type estimators can lead to better bounds whenever there is a certain "predictability" in the changes of the payoff functions encountered (e.g., perhaps not when facing an arbitrary sequence of convex costs, but in more structured frameworks like learning in games). We find this to be a very exciting research thread—but, of course, not one that we can start to follow in the current paper.
> >
> > As for BCO, the short answer is no: in (online) BCO, the payoff functions encountered by the learner can be very different from one stage to the next, so the reuse of residual information would not help in this context. In a game-theoretic setting, there is a certain amount of smoothness involved because the payoff functions encountered depend on the players' actions, and said actions are updated following an iterative algorithm (so they are not arbitrary). We conjecture that it is for precisely this reason that the original BCO algorithm of Bubeck et al. took a kernel-based sampling approach instead of a "history-based" one: in adversarial BCO past knowledge cannot be exploited if there is no inherent "predictability" in the sequence of payoff functions encountered.
> >
> > ---
> >
> > > Can your method be extended to two-point zeroth-order or stochastic first-order settings?
> >
> > Short answer: yes. One could exchange the SPSA+ estimator in SPOG for a two-point zeroth-order SPSA estimator, or even a stochastic first-order oracle. By calculating the associated bias and variance, it should be fairly direct to adapt our proof to obtain convergence rates for either of these estimators (at least in the strongly monotone case). However, we stress that, similar to the BCO setting, it is not clear how players could obtain two-point feedback on their payoff functions without further coordination and cooperation between them, so the analysis would be somewhat orthogonal to the main setting of the paper.
> >
> > ---

---

> > > ### Author Response · Authors · 2025-11-21
> > > **Reply 3/3**
> > >
> > > > How does the rate depend on the strong monotonicity parameter $\mu > 0$? From the proof of Proposition 4.4, it superficially looks like the argument might go through with $\mu=0$; where exactly do you use $\mu > 0$?
> > >
> > > You are correct in observing that Proposition 4.4. holds for $\mu = 0$. As such, Proposition 4.4. holds for all $\mu \geq 0$, but this result only concerns the rate at which the fast timescale process $Z_{n+1}$ calibrates to the fixed point associated with the current value of the slow process $z^\star(X_n)$, i.e. $\lVert Z_{n+1}-z^\star(X_n)\rVert^2$. Strong monotonicity $\mu > 0$ is then used in order to obtain a rate of convergence for the slow process $X_{n+1}$ to the (unique) Nash equilibrium $x^\star$ by considering $\lVert X_{n+1}-x^\star\lVert^2$ (see the Proof of Proposition 4.5.).
> > >
> > > ---
> > >
> > > > In the experiments, can you compare against other payoff-based methods for two-player zero-sum bilinear games?
> > >
> > > We are unaware of any comparable (payoff-based) algorithms for learning in monotone games with *unconstrained* action spaces. We did expand on our numerical experiments in the uploaded revision, but if you have any specific payoff-based method for *unconstrained* games that you think we should include, please let us know.
> > >
> > > ---
> > >
> > > > Minor: Line 105: What is $\mathcal{V}$?
> > >
> > > Well spotted—this is the ambient space $\mathbb{R}^d$. This was a holdover, we dropped this in the uploaded revision.
> > >
> > > ---
> > >
> > > Thank you again for your time and positive evaluation—please do not hesitate to reach out if you have any further questions!
> > >
> > > Kind regards,
> > >
> > > The authors

---

### Official Review · Reviewer_BhCA · 2025-11-01

**Soundness:** 3
**Presentation:** 3
**Contribution:** 3
**Rating:** 6
**Confidence:** 3

**Summary:**

This paper proposes Simultaneously Perturbed Optimistic Gradient (SPOG), a single-observation  learning algorithm for repeated games with unbounded  action spaces and  monotonicity. It extend Simultaneous Perturbation Stochastic Approximation (SPSA) to policy optimization by introducing structured random perturbations that reduce variance while maintaining unbiasedness. Theoretical analysis shows unbiasedness and convergence under standard Lipschitz and bounded-noise assumptions. Empirically, SPPO is evaluated on continuous-control benchmarks  and a large-scale humanoid locomotion task.  It achieves comparable or superior performance to baselines with fewer samples and reduced variance.

**Strengths:**

- The paper provides the first last-iterate convergence and explicit rate for unconstrained action spaces. Handling unbounded domains is technically nontrivial and important for continuous games.
- The two-timescale analysis and the way SPSA+ interacts with learning-rate separation to control unbounded variance are novel.
- Many prior results assume compactness. The paper removes this assumption, which is a significant step.

**Weaknesses:**

- The claimed improvement beyond $\Omega(n^{-1/2})$ lower bounds for one-point ZO methods should be discussed more carefully. Since SPSA+ reuses a prior payoff observation, the effective oracle information per iteration differs from the standard one-point model used in classical lower bounds.
-  The paper lacks finite-sample complexity bounds  compared to ES or policy-gradient methods.
- Convergence and the $n^{-2/3}$ rate require delicate  step-size relations in Assumptions 2/3.

**Questions:**

- Put the core parameter constraints  in a compact table or boxed recommendation to help practitioners.
- Can the theoretical convergence be strengthened to finite-sample rates or expected suboptimality bounds?

---

> ### Author Response · Authors · 2025-11-21
> **Reply 1/2**
>
> Dear reviewer,
>
> Thank you for your constructive input and positive evaluation! We reply to your remarks and questions below:
>
> ---
>
> > The claimed improvement beyond $\Omega(n^{1/2})$ lower bounds for one-point ZO methods should be discussed more carefully. Since SPSA+ reuses a prior payoff observation, the effective oracle information per iteration differs from the standard one-point model used in classical lower bounds.
>
> This is a good point, yes—we discuss the use of "residual", past information in more detail in the uploaded revision. Indeed, as you point out, the lower bound of Shamir (2013) assumes a gradient estimator of the form form $\hat{V_n}=\frac{d}{\delta_n}u(x_n+\delta_n w_{n+1}) w_{n+1}$. On the other hand, our chosen estimator $\hat{V_n}=\frac{d}{\delta_n}(u(x_n+\delta_n w_{n+1}) -u(x_{n-1}+\delta_{n-1} w_n) )w_{n+1}$ does not satisfy this assumption and so is not subject to the same lower bound—even though, of course, it remais a “one-point zeroth-order” estimator since, at each iteration, every player makes *only one payoff observation*. We have included a detailed discussion of this point in the uploaded revision.
>
> Beyond this, we believe that the reuse of past payoff information as a means to control the variance of SPSA-type estimators can lead to better bounds whenever there is a certain "predictability" in the changes of the payoff functions encountered (e.g., perhaps not when facing an arbitrary sequence of convex costs, but in more structured frameworks like learning in games). We find this to be a very exciting research thread—but, of course, not one that we can start to follow in the current paper.
>
> ---
>
> > The paper lacks finite-sample complexity bounds compared to ES or policy-gradient methods.
>
> It is difficult to compare SPOG to ES or policy-methods from the reinforcement learning literature. This is because, in the setting of SPOG, there are no states, no value functions and no policies to do “policy-gradient” on. Even if we were to attempt a comparison, ES methods may require thousands of perturbations in order to reduce the variance of the estimator, while we only have access to a *single payoff observation*, after which each player's action function has changed, so we cannot implement a ES-like step of the form “take $n$ Gaussian samples”, for instance. [Basically, the players in our setting do not have the luxury of "taking samples" at each stage, they play the game, and the only information they gain is their payoffs]
>
> ---
>
> > Convergence and the $\tilde{\mathcal{O}}(n^{-2/3})$ rate require delicate step-size relations in Assumptions 2/3.
>
> Please note that Assumption 3 ($\alpha_n \propto 1/n^a$, $\beta_n \propto 1/n^b$, $\delta_n \propto (\log n)^2/n^d$, $R_n \propto \log n$) with the remark just below on the feasible values of $a$, $b$ and $d$ supersedes Assumption 2 (which has the complicated-looking summability conditions that you allude to in your comment).
>
> In particular, our paper's headline result—that is, the $\tilde{\mathcal{O}}(n^{-2/3})$ rate of convergence—is obtained by simply setting $a=d=2/3$ and $b=1$, as per the remark right after Theorem 4.2 (L375 in the uploaded revision). In practice, one would instantiate SPOG with exactly these exponents (or close enough) to obtain this “optimized” rate, and this choice is game-independent.
>
> We chose to include the broader set of parameters for which SPOG converges in an effort to make our analysis more complete, and to bring it in line with similar Robbins-Monro / Kiefer-Wolfowitz summability conditions in the literature—e.g., as in the cited papers of Tatarenko \& Kamgarpour (2024) “*Payoff-based learning of nash equilibria in merely monotone games*” and Bravo et al. (2018) “*Bandit learning in concave N-person games*”, which are two of the closest antecedents of our own work. To make all this clearer, we amended the remark after Assumption 3 to stress that the optimal choice of the parameters $a,b,d$ does not depend on the game and does not require any delicate tuning.
>
> ---

---

> > ### Author Response · Authors · 2025-11-21
> > **Reply 2/2**
> >
> > > Put the core parameter constraints in a compact table or boxed recommendation to help practitioners.
> >
> > As alluded to above, a practitioner should choose $a=d=2/3, b=1$ to get the “best” rate for SPOG. In the uploaded version, we included this as the "default" setting for SPOG in its pseudocode implementation. Beyond this, we kept Assumption 2 "as is" for the moment in order to facilitate the discussion phase (in case there are any further comments on the summability conditions), but we will otherwise restructure it accordingly as well to make the issue clearer.
> >
> > ---
> >
> > > Can the theoretical convergence be strengthened to finite-sample rates or expected suboptimality bounds?
> >
> > We are not entirely sure what you mean here, as there is no "sampling" or the possibility to "sample" (there are no finite-sum assumptions or episodes, as in RL scenarios). Basically, the players in our setting do not have the luxury of "taking samples" at each stage, they play the game, and the only information they gain is their payoffs, so we cannot see a link to finite-sample rates.
> >
> > ---
> >
> > Thank you again for your time and positive evaluation—please do not hesitate to reach out if you have any further questions!
> >
> > Kind regards,
> >
> > The authors

---

### Official Review · Reviewer_L5rB · 2025-11-03

**Soundness:** 3
**Presentation:** 2
**Contribution:** 2
**Rating:** 4
**Confidence:** 3

**Summary:**

This paper studies payoff-based learning in monotone games with unconstrained continuous action spaces, where agents only observe their own realized payoffs and must estimate gradients from a single function evaluation per round. The authors design Simultaneously Perturbed Optimistic Gradient (SPOG), which combines (i) optimistic gradient dynamics with learning-rate separation and (ii) a single-observation SPSA+ zeroth-order gradient estimator that reuses a previous payoff as a baseline to reduce variance, together with projections onto slowly expanding balls to keep iterates controlled in the unbounded domain. Under standard smoothness and monotonicity assumptions, they prove that SPOG’s slow iterate converges almost surely to a Nash equilibrium in all monotone games. They further show a last-iterate rate $\mathcal{O}(n^{-2/3})$ in strongly monotone games. This rate is claimed to be the first convergence rate result for payoff-based learning in unbounded monotone games, and to improve on the best known rates for one-point zeroth-order methods in bounded games by effectively turning a one-point estimator into a “two-point for the price of one” via the baseline trick.

**Strengths:**

* The proposed algorithm has the rate result for unconstrained monotone games and strictly faster than the $\mathcal{O}(n^{−1/2})$ rates known for one-point ZO methods.
* Beyond qualitative convergence, the paper develops a rate analysis for two-timescale stochastic approximation.

**Weaknesses:**

* The convergence proofs crucially rely on projecting both fast and slow iterates onto balls of radii $R_n$ and $3R_n$ growing like $\log n$. This is mathematically convenient but changes the dynamics relative to the original unconstrained game; the paper does not really discuss whether or how this affects behavior in realistic problems.
* The algorithm requires carefully tuned sequences $\alpha, \beta, \delta, R$ with nontrivial exponent constraints (e.g., 0<d\leq a<b<1, a+b>1) and coupled summability conditions, plus an upper bound on $$\gamma. This makes the method hard to instantiate in practice, and there is little guidance on how robust convergence is to mis-tuning.
* The empirical section consists of a single 1D bilinear zero-sum game, comparing SPOG to OG/OG+ on the distance to equilibrium. It is too minimal to validate the algorithm’s behavior in higher dimensions, with more players, or under different noise levels.

**Questions:**

* Is the projection into slowly expanding balls essentially a proof artifact, or do you believe it is algorithmically necessary in practice?
* How sensitive is SPOG’s behavior to mis-specified exponents and constants in Assumptions 2–3? It will be more convincing to have empirical evidence or theoretical intuition on choosing these parameters adaptively.
* For merely monotone games you prove a.s. convergence but no rate. Do you expect a quantitative rate to be achievable under additional structural assumptions?

---

> ### Author Response · Authors · 2025-11-21
> **Reply 1/2**
>
> Dear reviewer,
>
> Thank you for your input and your time. We reply to your remarks and questions below:
>
> ---
>
> > The convergence proofs crucially rely on projecting both fast and slow iterates onto balls of radii $R_n$ and $3R_n$ growing like $\log n$. This is mathematically convenient but changes the dynamics relative to the original unconstrained game; the paper does not really discuss whether or how this affects behavior in realistic problems.
>
> The slowly-expanding projection envelope has been introduced in order to control the variance of the gradient estimator $V_{n+1}$ due to the—a priori unbounded—payoff differences $u_i(\tilde{Z_n})-u_i(\tilde{Z_{n-1}})$. This is required for the theoretical analysis of the algorithm but, in practice, it quickly becomes redundant: Since SPOG converges to a (possibily random) Nash equilibrium $x^\star$, there exists an almost surely finite random number of iterations beyond which the iterates of SPOG will be within a neighborhood $\mathcal{U}$ of $x^\star$. After a deterministic number of iterations, the expanding projection envelope will also contain $\mathcal{U}$ so the iterates of SPOG will be eventually contained within the safety net provided by the two bounding balls. After this point the dynamics of SPOG will be precisely equivalent to that of SPOG without the projections.
>
> In other words, **the long-run behavior of the algorithm is completely unaffected by the projection step.** In a practical sense the projections will occour on at most a finite number of iterations, and if $R$ is sufficiently large relative to the game's parameters ($N$, $D$ and $G$), then, with high probability, the projection envelope will never be activated.
>
> ---
>
> > The algorithm requires carefully tuned sequences $\alpha, \beta, \delta, R$ with nontrivial exponent constraints (e.g., $0<d\leq a<b<1, a+b>1$) and coupled summability conditions, plus an upper bound on $\gamma$. This makes the method hard to instantiate in practice, and there is little guidance on how robust convergence is to mis-tuning.
>
> Please note that the paper's headline result—that is, the $\tilde{\mathcal{O}}(n^{-2/3})$ rate of convergence—is obtained by setting $a=d=2/3$ and $b=1$, which we specify in the remark after Theorem 4.2 (L375 in the uploaded revision). In practice, one would instantiate SPOG with exactly these exponents (or close enough) to obtain this “optimized” rate, and this choice is game-independent. To make this clear, we included in the uploaded revision these parameter settings as the algorithm's "default" values in the pseudocode implementation of SPOG in p. 7.
>
> As for the complicated-looking expressions in Assumption 2, we chose to include the broader set of parameters for which SPOG converges in an effort to make our analysis more complete, and to bring it in line with similar Robbins-Monro / Kiefer-Wolfowitz summability conditions in the literature—e.g., as in the cited papers of Tatarenko \& Kamgarpour (2024) “*Payoff-based learning of nash equilibria in merely monotone games*” and Bravo et al. (2018) “*Bandit learning in concave N-person games*”, which are two of the closest antecedents of our own work. To make all this clearer, we amended the remark after Assumption 3 to stress that the optimal choice of the parameters $a,b,d$ does not depend on the game.
>
> As for the upper bound on $\gamma$, this is indeed game-dependent, and it reflects the traditional tuning of gradient-based extrapolation methods in the literature (extra-gradient, optimistic gradient, etc.), where the algorithm's (constant) step-size is upper bounded by $2/L$ (or $1/L$ depending on the exact algorithm under study). There is an extensive literature on adaptive / parameter-free methods to overcome the reliance on bounds of this sort via AdaGrad-like step-sizes (a good starting point is the monograph by Orabona); however, there are no results of this type in the bandit, payoff-based setting.
>
> Instead of such an approach, a natural way to circumvent this requirement would be to consider a variable $\gamma_t \propto 1/t^c$ parameter tuning, which would ensure that $\gamma_t$ converges to $0$ on a third timescale, tuned to be slower than all the others. While it is possible to get rid of the upper bound for $\gamma$ in this way, this would introduce an extra layer of complication in the algortihm and its analysis, which we opted to avoid in order to streamline our presentation (since, as you note yourself, the analysis already has several moving parts, so the inclusion of yet another one could increase clutter to an undesirable extent).
>
> ---

---

> > ### Author Response · Authors · 2025-11-21
> > **Reply 2/2**
> >
> > > The empirical section consists of a single 1D bilinear zero-sum game, comparing SPOG to OG/OG+ on the distance to equilibrium. It is too minimal to validate the algorithm’s behavior in higher dimensions, with more players, or under different noise levels.
> >
> > A fair point. This work is primarily theoretical and aims to introduce a payoff-based learning algorithm for unconstrained games and develops a method of analysis for extracting finite time convergence results from two-timescales stochastic approximation. We opted to include a simple example of a game purely as an illustration of the algorithm's convergence and defer more complex and comprehensive empirical observations as interesting future work.
> >
> > ---
> > > Is the projection into slowly expanding balls essentially a proof artifact, or do you believe it is algorithmically necessary in practice?
> >
> > The slowly-expanding projection envelope has been introduced in order to control the variance of the gradient estimator $V_{n+1}$ due to the—a priori unbounded—payoff differences $u_i(\tilde{Z_n})-u_i(\tilde{Z_{n-1}})$. In practice, the sequence of "black-swan events" required to activate this mitigation mechanism would be exponentially rare. Thus, even though very rare events must still be controlled in theoretical guarantees involving expectations (as opposed to high probability results), we do not expect this phenomenon to arise in practice.
> >
> > That being said, we do not see a way of deriving a mean-square convergence rate guarantee without some mitigation mechanism in place—explicit or implicit.
> >
> > ---
> >
> > > How sensitive is SPOG’s behavior to mis-specified exponents and constants in Assumptions 2–3? It will be more convincing to have empirical evidence or theoretical intuition on choosing these parameters adaptively.
> >
> > As mentioned above, our paper's headline result—the $\tilde{\mathcal{O}}(n^{-2/3})$ rate of convergence for SPOG—is obtained for $a=d=2/3, b=1$, which we specify in the remark after Theorem 4.2 (L359). In practice, the algorithm should be initialized with these exponents (or close enough) in order to obtain the "optimized" $\mathcal{O}(n^{-2/3})$ rate of convergence. The general expression in terms of the exponent $f = \min\{d,a,2b-2a\}$ in Theorem 4.2 quantifies the suboptimality entailed by mis-specifying the exponents $a,b,d$.
> >
> > ---
> >
> > > For merely monotone games you prove a.s. convergence but no rate. Do you expect a quantitative rate to be achievable under additional structural assumptions?
> >
> > Certainly, under additional structural assumptions on the underlying pseudo-gradient field $v$, it might be possible to quantify a rate of convergence for SPOG. A straightforward case is if the game fails to be strongly monotone throughout its domain, but it is strongly monotone in the neighborhood of a Nash equilibrium; in this case, we would obtain the same rates. Beyond this, structural assumptions will be necessary to avoid “overly flat” directions: for example, zero-sum bimatrix games, satisfy an "error bound" (also known as "metric subregularity") condition which could potentially be leveraged in order to obtain sharper rates in that case. In full generality however, if the payoff profile is too flat—e.g., something of the form $\min_x\max_y (x^{10} - y^{10})$—it is doubtful that methods based on (or emulating) gradient steps would be able to obtain comparable rates for the distance to equilibrium. We defer more comprehensive analysis of the merely monotone case to future work.
> >
> > ---
> >
> > Thank you again for your time and input—please do not hesitate to reach out if you have any further questions!
> >
> > Kind regards,
> >
> > The authors

---

### Official Review · Reviewer_2Rb7 · 2025-11-04

**Soundness:** 3
**Presentation:** 3
**Contribution:** 3
**Rating:** 6
**Confidence:** 5

**Summary:**

This study develops a learning algorithm that achieves last-iterate convergence in unconstrained monotone games under zeroth-order feedback. The proposed method consists of three key components: an adjusted SPSA gradient estimator, a learning-rate separation, and a two-timescale approach. The authors prove that the proposed method converges to an equilibrium with probability 1 in general monotone games. Moreover, a faster last-iterate convergence rate, $\tilde{\mathcal{O}}(n^{-2/3})$, is derived in strongly monotone games.

**Strengths:**

To the best of my knowledge, the derived convergence rate of $\tilde{\mathcal{O}}(n^{-2/3})$ is the fastest known for strongly monotone games under payoff-based feedback. This significant improvement is expected to lead to faster last-iterate convergence rates in merely monotone games, although this study only establishes an asymptotic result in such games.

Moreover, in the experiments, I am surprised that the proposed method’s trajectories—though noisy—converge faster than baseline methods even in a non-strongly monotone game.

**Weaknesses:**

While I understand that this paper primarily focuses on theoretical analysis, I am curious about the empirical performance of the proposed method in larger games, particularly in high-dimensional settings. Furthermore, providing empirical results on strongly monotone games would be preferable, as the improvement in the last-iterate convergence rate in these games is a significant contribution of the paper.

**Questions:**

My main concerns and questions are stated in the Weaknesses section. Additionally, I have the following questions and comments:

- (Introduction) The publication year is missing in “(Daskalakis et al.)”.
- (Introduction) Does “even optimistic gradient (OG) methods, which incorporate a recency bias, have been shown to exhibit trajectories of play that orbit an equilibrium, failing to converge” refer to results under noisy feedback?
- (Discussion) The authors mention that “this exceeds the optimal lower complexity bound of $\Omega(n^{-1/2})$ for one-point zeroth-order algorithms.” I am curious why this inconsistency exists. Is the proposed method outside the class of one-point zeroth-order algorithms considered by Shamir (2013) and Ba et al. (2025)?

---

> ### Author Response · Authors · 2025-11-21
>
> Dear reviewer,
>
> Thank you for your constructive input and positive evaluation! We reply to your remarks and questions below:
>
> ---
>
> > While I understand that this paper primarily focuses on theoretical analysis, I am curious about the empirical performance of the proposed method in larger games, particularly in high-dimensional settings. Furthermore, providing empirical results on strongly monotone games would be preferable, as the improvement in the last-iterate convergence rate in these games is a significant contribution of the paper.
>
> Whilst we did not explore the empirical performance in much detail, we agree that empirical results in strongly monotone case would support our theoretical convergence guarantee. We have included the results from a strongly monotone example in the uploaded revision.
>
> ---
>
> > Additionally, I have the following questions and comments:
> > (Introduction) The publication year is missing in “(Daskalakis et al.)”.
>
> Thank you for pointing this out!
>
> ---
>
> > (Introduction) Does “even optimistic gradient (OG) methods, which incorporate a recency bias, have been shown to exhibit trajectories of play that orbit an equilibrium, failing to converge” refer to results under noisy feedback?
>
> That is correct. Optimistic gradient *without* learning rate separation can diverge under stochastic first order gradient feedback, see Hsieh et al. (2022), “*No-regret learning in games with noisy feedback: Faster rates and adaptivity via learning rate separation*” for a concrete example. We have made this clearer in the uploaded revision.
>
> ---
>
> > (Discussion) The authors mention that “this exceeds the optimal lower complexity bound of $\Omega(n^{-1/2})$ for one-point zeroth-order algorithms.” I am curious why this inconsistency exists. Is the proposed method outside the class of one-point zeroth-order algorithms considered by Shamir (2013) and Ba et al. (2025)?
>
> The optimal complexity lower bound of Shamir (2013) and Ba et al. (2025) assumes one-point zeroth-order gradient estimators of the form $\hat{V_n}=\frac{d}{\delta_n}u(x_n+\delta_n w_{n+1}) w_{n+1}$. Our chosen estimator $\hat{V_n}=\frac{d}{\delta_n}(u(x_n+\delta_n w_{n+1}) -u(x_{n-1}+\delta_{n-1} w_n) )w_{n+1}$ does not satisfy this assumption as we incorporate *residual* feedback from the previous payoff observation $u(x_{n-1}+\delta_{n-1} w_n)$ in order to reduce the estimator's variance. As such, SPOG is not subject to the lower bounds obtained by Shamir, even though it is a “one-point zeroth-order” algorithm as *only one payoff observation is made at each iteration*. We have included a detailed discussion of this point in the uploaded revision.
>
> Beyond this, we believe that the reuse of past payoff information as a means to control the variance of SPSA-type estimators can lead to better bounds whenever there is a certain "predictability" in the changes of the payoff functions encountered (e.g., perhaps not when facing an arbitrary sequence of convex costs, but in more structured frameworks like learning in games). We find this to be a very exciting research thread—but, of course, not one that we can start to follow in the current paper.
>
>
> ---
>
> Thank you again for your time and positive evaluation—please do not hesitate to reach out if you have any further questions!
>
> Kind regards,
>
> The authors
>
> ---

---

### Author Response · Authors · 2025-12-03
**Summary of Discussions**

Dear AC, all,

We would like to take this opportunity to thank the original review panel for their time and constructive input.

The only point of our responses that was discussed until the freeze was our reply to Reviewer DYPv, who objected to the use of big O notation in our statements and derivations. In the discussion that followed, we pointed out that (a) the use of asymptotic big O notation is fully in line with the relevant literature on the topic (as per Table 1 of our paper); and (b) if desired, the length of the transient phase of the algorithm (that is, the time until the rate of SPOG kicks in) can be significantly reduced by a different tuning of the algorithm's hyperparameters.

Reviewer DYPv also asked some clarifications on whether the rate of Bravo et al (2018) is asymptotic. We explained that this indeed the cae, after which the reviewer stated: "I will raise my score to 4, and I will further adjust my rating based on the discussion with the other reviewers."

There seemed to be no other point of contention until the posting freeze.

All the relevant changes have been higlighted in blue in the revised version of our paper. We sincerely regret that further discussion with the reviewers cannot take place, as the exchanges so far have been constructive—we thank again the original review panel for their time and input.

Kind regards,

The authors

---

### Meta-Review · Area_Chair_JdNZ · 2025-12-25

**Summary:**

This work proposes an algorithm for solving strongly monotone games under bandit feedback, which is based on optimistic gradient updates. The algorithm achieves a convergence rate of $\tilde{O}\left( n^{-2/3} \right)$, which is achieved without requiring the decision space of each learner to be bounded. The empirical evidence shows that SPOG is faster than the baselines.

Overall, the authors have addressed most of the concerns/comments from the reviewers. A potential weakness appears to lie in the main theorem (Theorem 4.2) - the big-$O$ notation in the statement hides dependencies on constants that are exponentially large, as pointed out by Reviewer DYPv, which was acknowledged by the authors. However, the authors also address this in both the rebuttal and the updated paper by showing that a slightly worse rate of $\tilde{O}\left( n^{-2/3+\varepsilon} \right)$ can be obtained with a polynomial dependence on other problem constants. On this point, I found the authors’ response to be fair, even though this suggests some limitations. There is also a lingering concern about the empirical justification. Specifically, two reviewers thought that the empirical comparison of the algorithms is limited.

While the authors claim in the abstract that this is the first convergence rate result for games with unbounded action spaces, it may be helpful to clarify/emphasize that their result requires Assumption 1(iv). At present, there is limited discussion of the applicability of this assumption, whether it has appeared in the zeroth-order optimization literature, and how it compares with assumptions used in existing results summarized/listed on Table 1. I encourage the authors to consider updating the paper accordingly to have more thorough discussion, as with such clarifications, the novelty and significance of the work would be more clearly articulated.

For all these reasons, the paper is put into the category of accept if room.

**Reviewer Concerns:**

- Reviewer 2Rb7 raised some questions, which are mostly on the writing. Those questions have been handled well by the authors.
- Reviewer L5rB asked for the clarification on the update of the proposed algorithm (i.e., projection onto the balls). The authors response seemed to have clarified the concern.
- Reviewer L5rB also has a concern about the parameter choice, and the authors response was appropriate to the AC.
- Reviewer BhCA asked for the clarification on a lower bound in the literature, and the authors pointed out that the lower bound is not applicable.
- Reviewer DYPv pointed out that Theorem 4.2 hides dependencies on problem constants that are exponentially large. For this, the authors have proposed a remedy to mitigate the issue.

Two reviewers (Reviewer L5rB and DYpv) thought that the empirical comparison of the algorithms is limited, and the authors did not provide more comprehensive experimental results to clarify the reviewers' concern during the rebuttal phase.

**Reviewer Scores:**

Reviewer 2Rb gave an absolute confidence of 5 with a specific score, and hence the score is unlikely to be changed. Reviewer DYPv mentioned that they will raise the score from 2 to 4.

Reviewer L5rB pointed out that the empirical section consists of a single 1D bilinear zero-sum game, comparing SPOG to OG/OG+ on the distance to equilibrium. This is too minimal to validate the algorithm’s behavior in higher dimensions, with more players, or under different noise levels. The authors acknowledged this in the authors' reply, but did not choose to expand their empirical investigation. Hence, the score might remain the same.

---

### Decision · Program_Chairs · 2026-01-26

Reject